# HieraScaffold: Learning Compact Hierarchical Representations for Scalable 4D LiDAR Generation

**Zijie Wu** [1]   **Na Zhao** [1]

## Abstract

Outdoor LiDAR generation has shown strong potential for autonomous driving and large-scale 3D perception. However, existing approaches remain computationally intensive and primarily static, lacking explicit modeling of temporal dynamics. This limitation weakens spatiotemporal coherence and reduces the realism of 4D LiDAR generation. We propose a hierarchical recoupling generation framework that explicitly disentangles and reconstructs large-scale geometry and motion within a unified hierarchical structure. First, we design a multi-resolution feature scaffold that predicts time-correlated unsigned distance fields and spatial gradients, enabling hierarchical decomposition of 4D dynamics into static and motion-varying components. Next, to achieve compact yet expressive modeling, we introduce a neural contourlet representation that prunes redundant scaffolds into minimal directional bases, efficiently capturing essential geometric and motion cues. Finally, we progressively re-couple these hierarchical components to generate realistic and temporally coherent 4D LiDAR data. Extensive experiments demonstrate that our method outperforms baselines in both quality and consistency, achieving 3.3%, 25.0%, 17.8% improvements in FRD, MMD, and JSD, respectively, over the strong competitors, LiDMs and RangeLDM.

## 1. Introduction

LiDAR captures essential structural and dynamic information from the physical world, enabling a wide range of applications (Jia et al., 2025; Yang et al., 2026; Feng et al., 2025b) such as autonomous driving (Jiang et al., 2023; Wang et al., 2024; Wu et al., 2023), environmental perception (Wu et al., 2024; Sheng et al., 2025; 2022; Feng et al., 2025a), and motion planning (Hu et al., 2024b; Guo et al., 2025). However, collecting LiDAR data is both costly and labor-intensive, motivating recent efforts toward LiDAR data generation (Wu et al., 2023). In this paper, we propose a compact hierarchical representation that models realistic 4D spatio-temporal structures directly from raw LiDAR sequences, enabling scalable and coherent LiDAR scene generation.

To simulate real-world outdoor environments, range-view–based approaches (Caccia et al., 2019; Zyrianov et al., 2022; Ran et al., 2024; Pan et al., 2025) project a 360° LiDAR scan onto a 2D image encoding depth and intensity, enabling efficient generation at the frame level. However, such formulations are inherently constrained by the sparse and viewpoint-dependent sampling process of physical LiDAR sensors. Mapping irregular 3D measurements onto a fixed 2D grid discards geometric continuity and depth ordering, amplifies depth-dependent sparsity, and provides no explicit temporal correspondence across frames, *making it fundamentally difficult to model coherent and realistic 4D outdoor scenes*. To overcome these limitations, recent works model LiDAR (Tao et al., 2024; Zheng et al., 2024) directly in 3D space using point-based or rendered field representations, which better capture continuous geometry and realistic sensing effects. Despite their improved fidelity, these methods are typically computationally expensive and lack compact representations, posing significant challenges for efficient generative modeling. Gaussian splatting–based approaches (Hess et al., 2025; Kung et al., 2025) further improve re-simulation efficiency, yet their emphasis on accurate scene reconstruction requires modeling additional geometric and rendering dimensions, resulting in *increased computational and memory demands that limit scalability* in generation task. Moreover, unlike static 3D simulation, dynamic LiDAR scene generation imposes strict spatio-temporal coherence requirements. While methods such as LiDARCrafter (Liang et al., 2025) achieve fine-grained text-driven control over 4D LiDAR generation via explicit structural modeling, their formulation primarily targets localized dynamic regions and remains *limited in its ability to scale to generation over large, unbounded outdoor environments*.

To this end, we propose HieraScaffold, a hierarchical recou-

[1]Singapore University of Technology and Design, Singapore. Correspondence to: Na Zhao <na_zhao@sutd.edu.sg>.

*Proceedings of the 43rd International Conference on Machine Learning*, Seoul, South Korea. PMLR 306, 2026. Copyright 2026 by the author(s).

pling generative framework that progressively synthesizes large-scale dynamic LiDAR scenes from compact spatio-temporal representations. We first introduce a 4D dynamic scaffold that jointly models spatial geometry and temporal evolution directly from raw LiDAR sequences. Equipped with multi-resolution feature nodes, the scaffold learns continuous unsigned distance fields and their gradients from sparse point clouds, enabling accurate and high-quality reconstruction of dynamic scenes. To achieve an efficient generative representation, we develop a multi-stage neural contourlet-based decoupling strategy that removes redundancy in the frequency domain, yielding a compact latent scaffold. The decoupled representation is further factorized into directional sub-components, facilitating fine-grained structural modeling with reduced computational overhead. Finally, we propose a hierarchical recoupling generation process that progressively restores the scalable 4D scene from sub-scaffolds, reassembling the decomposed spatio-temporal information to preserve spatial and temporal coherence. In summary, our contributions are:

- We propose a dynamic scaffold representation that captures continuous spatiotemporal geometry through unsigned distance based LiDAR modeling.
- We develop a neural contourlet decoupling and recoupling mechanism that yields compact, direction-aware latent representations, enabling scalable and coherent 4D LiDAR generation.
- We introduce a Hierarchical Recoupling Generation framework that unifies geometry–motion disentanglement and scalable hierarchical 4D LiDAR generation.

## 2. Related work

**LiDAR Scene Generation:** LiDAR (Nakashima et al., 2025) captures geometry in complex environments, but its sparse and irregular sampling makes direct point cloud synthesis difficult, especially for large-scale dynamic scenes. Rangeview-based methods (Ran et al., 2024; Hu et al., 2024a; Zyrianov et al., 2022) map point clouds to panoramic depth images to build spatial structure in 2D, while UltraLiDAR (Xiong et al., 2023) uses a compact grid representation for efficient 3D modeling. These methods process scenes independently, overlooking spatio-temporal dependencies required for continuous and globally coherent dynamics.

**LiDAR Point Cloud Re-simulation:** LiDAR re-simulation has been widely explored to recover scene geometry and dynamics with high fidelity. NeRF-based methods such as LiDAR-NeRF (Tao et al., 2024), NeRF-LiDAR (Zhang et al., 2024), and NFL (Huang et al., 2023b) reconstruct scenes through neural radiance fields that capture fine-grained spatial structure and realistic sensor responses. To further enhance spatio-temporal modeling, LiDAR4D (Zheng et al., 2024) introduces additional temporal planes to form a high-

precision 4D hybrid representation, improving dynamic reconstruction capability. Gaussian Splatting methods (Chen et al., 2024b; Hess et al., 2025; Kung et al., 2025) are introduced to improve the efficiency of LiDAR re-simulation, but they still involve computational overhead during generation. These methods primarily focus on faithful scene re-rendering, achieving detailed geometry recovery but suffering from high computational and memory costs, which hinder scalability to large-scale dynamic environments.

**Dynamic Scene Generation:** With the rapid development of generative models (Mirza & Osindero, 2014; Rombach et al., 2022; Yang et al., 2025), particularly denoising diffusion models (Ho et al., 2020; Sohl-Dickstein et al., 2015) such as latent diffusion (Rombach et al., 2022) and DiT (Mo et al., 2024), dynamic scene generation has drawn increasing attention. DynamicCity (Bian et al., 2025) introduces a diffusion-in-transformer (DiT) framework with a Hex-Plane (Cao & Johnson, 2023) representation to generate 4D dynamic occupancy. DreamScene4D (Chu et al., 2024) constructs dynamic scenes by factorizing motion from multi-frame video inputs. LidarDM (Zyrianov et al., 2025) builds a scalable 3D dynamic background combined with object-level motion planning to enable dynamic LiDAR generation. LiDARCrafter (Liang et al., 2025) decomposes generation into a layout-to-scene and scene-to-sequence pipeline to learn dynamic patterns from LiDAR sequences, but its reliance on observed dynamics without explicit geometric grounding or hierarchy limits long-range consistency and scalability in 4D LiDAR generation.

## 3. Method

Generating large-scale LiDAR scenes requires building compact and structured representations from raw point clouds while designing a modeling pipeline capable of handling complex spatial–temporal content. As shown in Fig. 1, we first formulate geometric dynamics as a scaffold of unsigned distance and gradient flow modeling in Sec. 3.1, enabling structured decoupling of static and dynamic components. Sec. 3.2 then to further improvement by introducing a compact neural contourlet representation with directional decomposition. Building on these parts, Sec. 3.3 presents a hierarchical recoupling generation framework that synthesizes compact scaffold elements and recouples them to reconstruct scalable and coherent 4D LiDAR scenes.

### 3.1. Reconstructing Unsigned Distance Scaffolds from LiDAR sequences

LiDAR scans provide sparse geometric observations that capture the overall structure of a scene but lack dense surface details between sampled points. To infer the missing geometry, we first construct a *scaffold feature representation* from sparse LiDAR inputs to reason about continuous

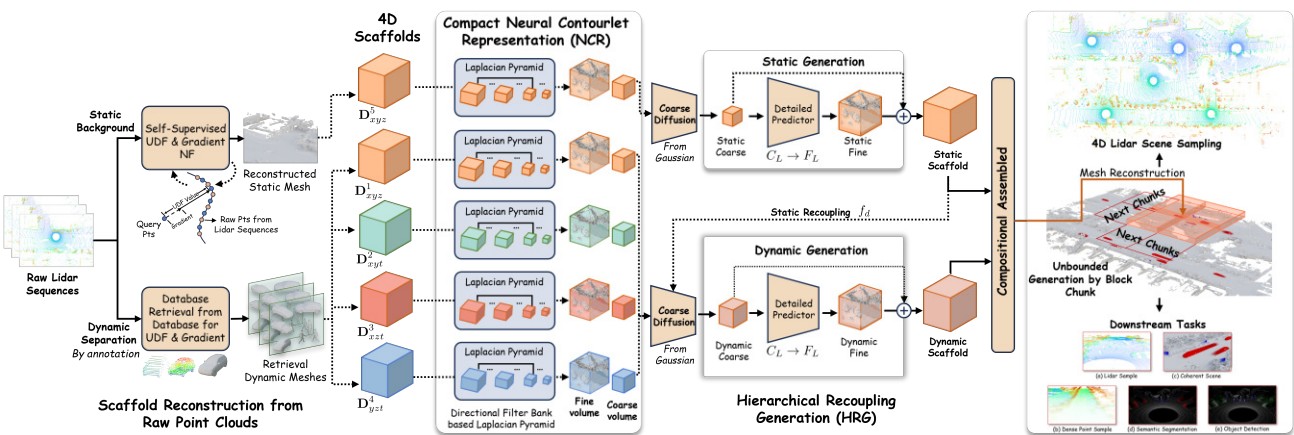

*Figure 1.* Overview of the proposed framework. Left: We create the 4D scaffold from raw lidar sequences. Middle: Decompose the scaffolds for efficient representation. Right: Hierarchical recoupling generation for 4D lidar scene generation.

surface geometry. Specifically, given a set of raw LiDAR sequences $\mathcal{P} = \{p_1, p_2, \cdots, p_i \mid i \in [1, M]\}$, we firstly decompose the scene $p_i$ into a static background $o^s$ and a dynamic foreground $o^d$ by labelled annotations, which contains dynamic objects such as vehicles and pedestrians. $o^d, o^s$ are then processed independently to construct separate feature cubes of scaffolds $\mathcal{D} = \mathbf{D}^{1,2,3,4}, \mathbf{D}^5$, following K-planes (Fridovich-Keil et al., 2023) decomposition.

**Self-supervised Neural Field (NF) learning for Static Scaffold.** Since ground-truth meshes are unavailable for large-scale outdoor backgrounds, we employ a self-supervised neural field learning method to recover the zero-level surface and perform *surface reasoning* by aligning local neighborhoods through an Unsigned Distance Field (UDF) for the static component. Conversely, for the dynamic components, we query object meshes from a pre-defined database to directly supervise the UDF modeling. As shown in Fig. 2, we introduce a transformer-based neural network to learn an adjacency-aware UDF neural field around the LiDAR point cloud, formulated as NF : $\mathbb{R}^3 \rightarrow \mathbb{R}^d$, where $d = 6$ represents the concatenation of the unsigned distance value $v \in \mathbb{R}^3$ and its corresponding gradient $g \in \mathbb{R}^3$.

Following recent implicit surface learning strategies (Zhou et al., 2024a; Chen et al., 2024a), we sample query points $q_i \in \mathcal{Q}$ around raw LiDAR points, where $\mathcal{Q}$ denotes a set of spatial samples near potential surface locations. We build a learnable 3D feature cubic scaffold $\mathbf{D}^5$ for static modeling, For each node of the scaffold, we construct $L = 3$ levels of learnable multi-resolution feature grids. For any query point $q_i$, feature vectors are extracted via trilinear interpolation from each layer and concatenated to form the final input $f(q_i)$, with grid parameters jointly optimized alongside the reconstruction loss. Then, neural field (NF) to infer both the unsigned distance $v_i$ and the gradient flow $g_i$ for each query point $q_i$. The gradient $g_i = \nabla \text{NF}(q_i)$ represents the local direction of maximum unsigned distance

variation in 3D space, which indicates the outward direction from the implicit surface. By moving the query point $q_i$ along the inverse direction of $g_i$ with a stride proportional to the predicted distance $v_i$, we iteratively guide the query toward the underlying surface by $s_i = q_i - v_i \cdot g_i / ||g_i||$. This geometric moving operation serves as a self-supervised constraint that enforces the predicted distance and gradient to be consistent with the surface structure encoded by the LiDAR samples. As for each query point $q_i \in \mathcal{Q}$, where $\mathcal{Q}$ denotes a set of spatial samples around the LiDAR points $\mathcal{P}$, the neural field predicts its unsigned distance and gradient as $[v_i, g_i] = \text{NF}\big(\text{PE}(q_i), f(q_i)\big)$, where $\text{PE}(\cdot)$ is the positional encoding of the query coordinate and $f(\cdot)$ denotes the query feature extracted from the local neighborhood.

To optimize the neural field, at this time, we now can apply a Chamfer-based consistency loss between the moved query points $s_i$ and the original LiDAR points $p_j$ to guide the neural field learning as:

$$\mathcal{L}_{\text{pts}} = \frac{1}{|\mathcal{S}|} \sum_{s_i \in \mathcal{S}} \min_{p_j \in \mathcal{P}} \|s_i - p_j\|_2^2 + \frac{1}{|\mathcal{P}|} \sum_{p_j \in \mathcal{P}} \min_{s_i \in \mathcal{S}} \|p_j - s_i\|_2^2.$$

(1)

Consequently, we obtain the static scaffold $\mathbf{D}^5$. In contrast to the static background, the dynamic component requires explicit temporal modeling. Drawing inspiration from K-Planes, we construct four spatiotemporal cubic scaffolds to factorize and represent the continuous geometry and motion of dynamic objects.

**Dynamic Unsigned Distance Scaffolds.** Leveraging the explicit decoupling of dynamic objects $o^d$ via dataset annotations, we retrieve corresponding meshes from the database to serve as geometric priors. We then compute the ground-truth Unsigned Distance Field (UDF) values from these meshes and encode this structural information into the 4D spatiotemporal scaffolds $\mathbf{D}^{1-4}$. Prior 4D methods like K-Plane (Fridovich-Keil et al., 2023) and HexPlane (Cao & Johnson, 2023) rely on multiple 2D latent planes and can-

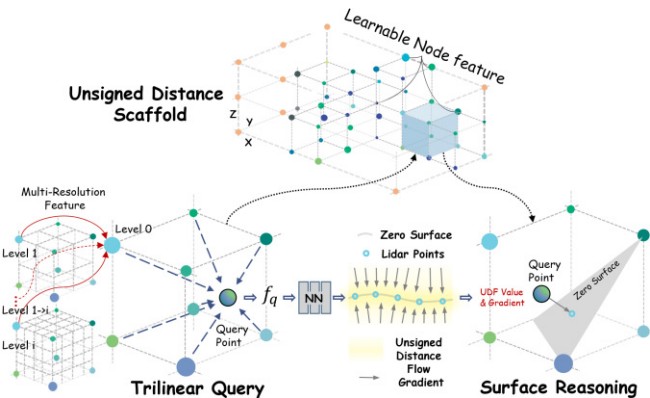

*Figure 2.* Scaffolds reconstruction from raw point cloud for surface reasoning.

not maintain these hierarchical relationships. Following LiDAR4D, we instead build a volumetric feature space using learnable keypoint-based nodes. Trilinear interpolation within this scaffold provides adaptive resolution while avoiding cubic computational growth, enabling scalable modeling of dynamic scenes with geometry and temporal coherence.

In summary, the proposed scaffold factorization employs five volumetric bases. We denote the purely spatial cubic volume as $\mathbf{D}^1_{xyz}$, and the three spatio-temporal cubes for dynamic modeling as $\mathbf{D}^2_{xyt}$, $\mathbf{D}^3_{xzt}$, and $\mathbf{D}^4_{yzt}$. Since the static background is decoupled as an independent component without temporal variation, we represent it using $\mathbf{D}^5_{xyz}$. We separate static and dynamic parts because the static region is relatively uniform, while dynamic areas are sparse and irregular, allowing each to be modeled more efficiently. For any query in 4D coordinates $q_i = \{x, y, z, \tau\}$, its feature is obtained by trilinear interpolation over the surrounding scaffold nodes from $\mathcal{D} = \mathbf{D}^{1-4}, \mathbf{D}^5$.

$$f(q_i) = \sum_{a,b,c \in \{0,1\}} w_{abc}(q_i)\, \mathcal{D}(x_a, y_b, z_c, \tau), \quad (2)$$

where $w_{abc}(q_i)$ denotes the normalized interpolation weights based on the relative distances between $q_i$ and its eight neighboring nodes. Each node consists of a concatenated multiresolution feature. This interpolation enables continuous feature sampling within discrete scaffold volumes, ensuring smooth spatial–temporal learning. This formulation allows the network to capture local geometric continuity and build structured representations from raw point clouds for generation.

*Mesh and LiDAR Sampling:* During sampling from the well-established scaffold $\mathcal{D}$, we first initialize a spatiotemporal UDF query grid of points to capture the dynamic components. Subsequently, we utilize the feature $f(q)$ derived from the transformer-based Neural Field (Eq. 2) to decode the UDF and gradient fields. Following the DCUDF (Hou et al., 2023) framework, we generate the underlying surface mesh using the double recovering strategy and finally apply raycasting to synthesize the LiDAR point clouds. Note that

the dynamic and static scaffold queries are conducted separately and then composited to form the complete scene, as shown in Fig. 1(right).

### 3.2. Compact Neural Contourlet Representation for 4D Scene Modeling

We represent sparse LiDAR scans using the scaffold set $\mathcal{D} = \{\mathbf{D}^{1-5}\}$, where each volumetric tensor captures spatial–temporal structure at high resolution. However, large outdoor scenes contain extensive empty regions and strong geometric regularities, leading to significant redundancy across both space and time. Directly using full scaffolds for generation is prohibitively expensive due to their high dimensionality. Therefore, we introduce a Neural Contourlet Representation (NCR) that maps scaffold features into a structured, frequency space to reduce redundancy. See suppl. for more detials.

**Redundancy minimization:** Frequency-domain methods such as wavelet decomposition offer efficient representations; for instance, WaveGen (Hui et al., 2022) shows that removing 97% of coefficients changes reconstruction by only 2.8%, reveal the potential for compact transformation. Inspired by this, we design a learnable transformation suited for LiDAR scenes, where large outdoor environments contain strong directional patterns like roads and walls that support anisotropic, compact encoding. We first to revisit the contourlet transform and adopt its core idea for NCR. Following (Higaki et al., 2008), NCR applies a 3D Laplacian pyramid with a directional filter bank (Fig. 4(a)):

$$L_J(x,y,z) + \sum_{j=1}^{J} \sum_{k=1}^{K_j} B_{j,k}(x,y,z) = \phi_\alpha\{I(x,y,z)\},$$

$$\hat{I}(x,y,z) = \phi_\gamma\{L_J(x,y,z) + \sum_{j=1}^{J} \sum_{k=1}^{K_j} \tilde{B}_{j,k}(x,y,z)\}. \quad (3)$$

where $L_J$ is the low-frequency component, $B_{j,k}$ are directional bands and $\phi_\alpha$ is the contourlet decompositor. The signal is reconstructed by the inverse transform $\phi_\gamma$. Following (Zhou et al., 2024b), we employ a learnable wavelet filters for the basic frequency transform. Building upon the above, for each cubic scaffold $\mathcal{D}$, we first apply a Laplacian pyramid decomposition in cubic space, denoted as $\{l_i \mid i \in [0, L]\}$, where $L$ is the total number of pyramid levels. (1) At level $i \in [0, L)$, the redundancy minimization process is defined:

$$l_{i+1} + R_{i+1} = \phi_\alpha\big(I_i(x,y,z)\big), \quad (4)$$

where $\phi_E(\cdot)$ denotes the encoding operator that separates the essential low-frequency structure $l_{i+1}$ from the redundant residual component $R_{i+1}$. During the reverse process, we discard the redundancy term $R_{i+1}$ to maintain compactness and retain only the structural representation $l_{i+1}$ for

hierarchical reconstruction:

$$\hat{I}_i(x, y, z) = \phi_\gamma(l_{i+1}), \qquad (5)$$

The objective is to minimize the difference between $I_{i+1}$ and its reconstruction $\hat{I}_{i+1}$ by optimizing the filter parameters $\alpha$ and $\gamma$ by $\min_{\alpha,\gamma} \mathcal{L}_{\text{MSE}}(I_{i+1}, \hat{I}_{i+1})$. where $\mathcal{L}_{\text{MSE}}$ denotes the mean squared error loss measuring the reconstruction fidelity between the original and recovered scaffold.

**Core Detangling:** (2) At the final level $i = L$, we assume that redundancy has been largely removed through the previous levels. Therefore, we apply the contourlet transform once more to decompose the remaining structural information: $C_L, F_L = \phi_\alpha(l_L)$, where $C_L$ denotes the low-frequency coarse volume and $F_L$ represents the high-frequency detailed volume that captures geometric variations. In practice, the raw $F_L$ tensor is considerably larger than typical feature maps; hence, we apply critical sampling (Do & Vetterli, 2005), which has been demonstrated to preserve information under lossless reconstruction. During the reverse process, both $C_L$ and $F_L$ are considered jointly, rather than discarding components as in the redundancy minimization stage. This ensures complete recovery of the hierarchical representation while maintaining compactness. The resulting feature set forms the most compact latent representation for generation tasks.

To further refine the representation quality, we joint-tuning the contourlet filter parameters while freezing the transformer-based NF parameters $\theta_{\text{NF}}$, and optimize:

$$\min_{\alpha,\gamma,\theta}; \mathcal{L}_{\text{fine}} = \mathcal{L}_{\theta_{NF}}^{\text{MSE}}(I, \hat{I}) + \lambda_c \mathcal{L}_{\phi_\alpha,\phi_\gamma}^{\text{tuning}}(U, \hat{U}, \theta_{\text{NF}}), \quad (6)$$

where $\mathcal{L}_{\text{MSE}}$ measures the reconstruction fidelity, and $U, \hat{U}$ are GT and query udf values. $\mathcal{L}_{\text{tuning}}$ joint fine-tuning learnable NCR parameters in the latent feature domain while keeping $\theta_{\text{NF}}$ frozen.

### 3.3. Hierarchical Recoupling Generation

Building upon the compact contourlet representation $C_L^{1-5}$ obtained from the previous stage, we now reconsider how to effectively generate spatio-temporally continuous 4D LiDAR scenes while achieving scalability for large outdoor environments. We design a Hierarchical Recoupling Generation (HRG) scheme that progressively recouple the decomposed components as shown in Fig. 1(right).

1) **Hierarchical Coarse2Fine Prediction** $\mathcal{G}$. It first generate starting from the scaffold level, we do not generate both the coarse volume $C_L$ and the fine detail volume $F_L$, since the compact coarse representation already contains most scene structure. Instead, we apply a detail predictor network that refines geometry and motion conditioned on the coarse feature. It follows the same architecture as the coarse generator

but adds extended convolutions along each spatial axis to handle the larger coefficient volume. The network learns a mapping: $f : C_L \rightarrow F_L$, predicting detailed coefficients from the coarse representation. Each coarse2fine prediction $\mathcal{G}$ chain of scaffold is defined:

$$p_{\beta_C}(C_L|\mathcal{C}) = p[\mathbf{x}_{C(T)}] \prod_{t=1}^{T} p_{\beta_C}(\mathbf{x}_{C(t-1)}|\mathbf{x}_{C(t)}, \mathcal{C})$$

$$p_{\beta_F}(F_L|C_L) = p[\mathbf{x}_{F(T)}] \prod_{t=1}^{T} p_{\beta_F}(\mathbf{x}_{F(t-1)}|\mathbf{x}_{F(t)}, C_L)$$
$$(7)$$

where $p(\mathbf{x}_T)$ is the sample from Gaussian, $\beta_C, \beta_F$ are the parameters of coarse diffusion, fine predictor and share the same architecture of a diffusion transformer based generator with different number of parameters. $\mathcal{C}$ is the condition input. As we get $F_L, C_L$, we can reconstruct the complete scaffold reversing contourlet pyramid:

$$\mathcal{D} = \prod_{i=L-1,\cdots 1} \phi_\gamma[i](\phi_\gamma[i+1], \varnothing) \cdot \phi_\gamma(C_L, F_L) \quad (8)$$

where $\varnothing$ indicates zero-padding applied during reconstruction to take place the redundancy input for reverse NCR and ensure consistent resolution alignment.

2) **Progressive Recoupling Generation:** This stage restores global spatial–temporal coherence from compact latent features. Our hierarchical scaffold representation models static and dynamic components separately and organizes each into independent scaffold blocks $\mathcal{D}$. Each block $\mathbf{D}$ is then recovered by generator $\mathcal{G}$ from a coarse latent and its corresponding fine details. Unlike existing diffusion models that generate all components as a single dense tensor, we first synthesize the static geometry and then infer dynamic objects conditioned on it. This produces a structured mapping that captures static–dynamic dependencies, enabling more realistic and coherent 4D LiDAR generation:

$$\mathcal{D} = \mathcal{G}_d(\mathbf{D}^{1,2,3,4}|\mathcal{M}(\mathcal{G}_s(\mathbf{D}^5)), \mathcal{C}) \cdot \mathcal{G}_s(\mathbf{D}^5|\mathcal{C}) \quad (9)$$

where $\mathcal{G}_s(\mathbf{D}^5)$ is the static generation, $\mathcal{C}$ denotes the condition input, and $\mathcal{M}$ is the dynamic-static mapping that is defined as:

$$\mathcal{M}(Q_i = f_{\mathbf{D}^{1:4}}, K_i \& V_i = f_{\mathbf{D}^5}) = \text{sfm.}[\sum_{i=1}^{\mathcal{Z}} \odot(\frac{Q_i K_i^{'}}{\sqrt{\mathcal{D}}})] \cdot V_i$$
$$(10)$$

where $Q_i$ is an element of the query static embedding, the condition keys $K_i, V_i$ are the layer features of $\mathcal{G}_d$, and $f_d$ denotes the updated feature correlated with static condition. $\odot$ denotes the Hadamard Product along the $\mathcal{Z}$-axis. We treat all $\mathcal{Z}$-axis slices as a single unit and compute the mapping only along the horizontal dimensions.

3) **Block-Chunk Generation:** We now define our large scale of dynamic scene generation. Most existing scene

*Table 1.* Reconstruction comparison of dynamic LiDAR scene on KITTI360 and Waymo dataset.

| Dataset | Methods | Venue | Point Cloud | | Range Image | |
|---|---|---|---|---|---|---|
| | | | CD ($\downarrow$) | F-Score ($\uparrow$) | RMSE ($\downarrow$) | SSIM ($\uparrow$) |
| KITTI360 dataset | NSKR (Xiong et al., 2023) | CVPR2023 | 0.5910 | 0.8419 | 5.8729 | 0.6937 |
| | K-planes (Hu et al., 2024a) | CVPR2023 | 0.1244 | 0.9040 | 4.2985 | 0.6628 |
| | LidarDM (Zyrianov et al., 2025) | ICRA2025 | 0.1584 | 0.9003 | 4.1261 | 0.6409 |
| | Ours | *present* | **0.0946** | **0.9214** | **3.0158** | **0.7581** |
| Waymo Open dataset | NSKR (Xiong et al., 2023) | CVPR2023 | 6.8796 | 0.6934 | 10.5315 | 0.5425 |
| | K-planes (Hu et al., 2024a) | CVPR2023 | 0.4107 | 0.8312 | 7.9424 | 0.7133 |
| | LidarDM (Zyrianov et al., 2025) | CVPR2025 | 0.9245 | 0.7758 | 7.7392 | 0.7048 |
| | Ours | *present* | **0.3214** | **0.8521** | **6.2405** | **0.7754** |

*Table 2.* Comparison of generation results on the KITTI360 (Behley et al., 2019) and Waymo open (Fong et al., 2022) datasets

| Method | Venue | KITTI360 dataset | | | Waymo Open dataset | | |
|---|---|---|---|---|---|---|---|
| | | FRD ($\downarrow$) | $\text{MMD}_{\text{BEV}}$ ($\downarrow$) | $\text{JSD}_{\text{BEV}}$ ($\downarrow$) | FRD ($\downarrow$) | $\text{MMD}_{\text{BEV}}$ ($\downarrow$) | $\text{JSD}_{\text{BEV}}$ ($\downarrow$) |
| LiDARGen (Zyrianov et al., 2022) | ECCV 2022 | 2040.1 | $3.87 \times 10^{-4}$ | 0.067 | 2567.4 | $2.23 \times 10^{-3}$ | 0.198 |
| UltraLiDAR (Xiong et al., 2023) | CVPR 2023 | – | $1.96 \times 10^{-4}$ | 0.071 | 2038.6 | $1.25 \times 10^{-3}$ | 0.211 |
| LiDMs (Ran et al., 2024) | CVPR 2024 | 1012.5 | $2.64 \times 10^{-5}$ | 0.049 | 1251.3 | $1.37 \times 10^{-4}$ | 0.129 |
| RangeLDM (Hu et al., 2024a) | ECCV 2024 | 1074.9 | $3.07 \times 10^{-5}$ | 0.045 | 1189.0 | $1.58 \times 10^{-4}$ | 0.138 |
| LidarDM (Zyrianov et al., 2025) | ICRA 2025 | 1423.2 | $8.74 \times 10^{-5}$ | 0.055 | 1934.3 | $1.07 \times 10^{-3}$ | 0.154 |
| R2Flow (Nakashima et al., 2025) | ICRA 2025 | 1047.3 | $2.87 \times 10^{-5}$ | 0.041 | 1321.4 | $1.41 \times 10^{-4}$ | 0.124 |
| Ours | *present* | **979.3** | $\mathbf{1.98 \times 10^{-5}}$ | **0.037** | **1087.4** | $\mathbf{9.35 \times 10^{-5}}$ | **0.059** |

generation methods represent the entire large-scale scene as a single unified latent volume and perform outward outpainting through iterative diffusion (Lugmayr et al., 2022). This strategy requires repeated denoising steps for each boundary region, resulting in significant computational overhead. Instead, inspired by the idea of Chunk-VAE (Lee et al., 2025), we divide the scene into multiple *block chunks* along the horizontal axes, allowing each block to be modeled and generated independently while maintaining inter-block coherence, as shown in Fig. 1(Right). This block-chunk formulation enables efficient localized generation and parallel processing, while ensuring seamless spatial transitions between neighboring chunks within the scaffold pyramid for large scale of LiDAR scene generation.

## 4. Experiment

**Datasets:** We train and evaluate our method on two outdoor LiDAR datasets: KITTI-360 dataset (Behley et al., 2019; Geiger et al., 2013) and Waymo Open Dataset (Sun et al., 2020).

**Baselines:** For reconstruction evaluation, we compare with NSKR (Huang et al., 2023a) (static *3D-only*), LidarDM (Zyrianov et al., 2025) (partial dynamic), and K-Planes (Fridovich-Keil et al., 2023) (full *4D*), covering a spectrum of spatial–temporal modeling capacities. For generation evaluation, we benchmark against diffusion- and transformer-based lidar generators including LiDMs (Ran et al., 2024), UltraLiDAR (Xiong et al., 2023),

RangeLDM (Hu et al., 2024a), and LidarDM (Zyrianov et al., 2025). For scene graph-based controllable generation, we compare our method with LiDARCrafter (Liang et al., 2025). To ensure a fair comparison, we adopt their scene graph-based text-to-layout generation pipeline as the condition input.

**Evaluation Metrics:** We evaluate reconstruction quality using Chamfer Distance and F-score (5 cm). For generative performance, we report FRD, $\text{MMD}_{\text{BEV}}$, and $\text{JSD}_{\text{BEV}}$ to assess geometric accuracy, structural consistency, and distributional diversity of the synthesized 4D LiDAR scenes. See more details in Suppl.

### 4.1. 4D Dynamic LiDAR Comparisons

**Reconstruction:** We first validate the reconstruction quality of our scaffold representation on the KITTI-360 and Waymo datasets. Tab. 1 presents a detailed comparison between our method and existing baselines on both point cloud and range image domains. For fair comparison, we adopt the same ray-drop simulation module as Lidar-DM. Compared with the 3D-only NSKR, methods incorporating 4D spatial–temporal modeling show clear improvements in reconstruction accuracy. However, Lidar-DM separates the dynamic object branch from the static encoder, leading to inconsistent representations and higher reconstruction error. Our method achieves the best overall performance, reducing Chamfer Distance and improving F-Score by 24.0%/1.9% on KITTI-360 and 21.7%/2.1% on Waymo, respectively,

surpassing the nearest competitor K-Planes across all metrics. These results demonstrate the effectiveness of our unified hierarchical scaffold in producing more accurate and coherent LiDAR reconstructions.

**Generation:** We further evaluate the quality of our generated dynamic LiDAR scenes through two complementary analyses. *(1) 3D LiDAR Point Cloud Evaluation:* We assess realism by rendering synthetic point clouds under the same sensor configuration as the datasets, matching elevation angles, azimuth spacing, and field of view. As reported in Tab. 2, our method consistently achieves the best performance. On KITTI-360, it surpasses LiDMs by 3.3% in FRD, 25.0% in $MMD_{BEV}$, and 24.5% in $JSD_{BEV}$, and outperforms RangeLDM by 8.9%, 35.5%, and 17.8%, respectively. On Waymo Open, the gains are even larger, reducing JSD by 54.3% over LiDMs, indicating a closer match to real-world distributions. These findings confirm that our hierarchical grounded generation framework produces LiDAR data with strong geometric fidelity and distributional consistency, generalizing well across diverse outdoor environments.

*Table 3.* Comparison of temporal consistency on the Waymo dataset. Outlier percentage distance threshold is 0.5m.

| Method | Total ICP Energy[m] | Average ICP Energy | Outlier Percentage | Chamfer Distance [m] |
|---|---|---|---|---|
| Seq. Diffu. | 3616.58 | 0.078 | 20.56% | 0.39 |
| LidarDM | 916.94 | 0.014 | 7.12% | 0.17 |
| Ours | **735.40** | **0.009** | **5.38%** | **0.11** |

*(2) 4D Temporal Coherency.* We further evaluate the temporal consistency of our 4D dynamic LiDAR generation, following LidarDM (Zyrianov et al., 2025), where sequence diffusion is implemented using a video diffusion model. To measure temporal alignment between consecutive frames, we compute the average Iterative Closest Point (ICP) energy, where lower values indicate better spatial–temporal coherence. This metric reflects the geometric stability across time, smaller average energy implies smoother motion and stronger temporal consistency. As shown in Tab. 3, sequence diffusion fails to capture fine-grained spatial–temporal relations, resulting in high ICP energy and discontinuous motion. LidarDM improves temporal planning for dynamic objects but overlooks consistency with static background regions, leading to residual misalignment. In contrast, our method achieves significantly lower total and average ICP energy, reducing outliers by 24.4% and Chamfer Distance by 35.3% compared with LidarDM. These results confirm that our unified 4D recoupling framework ensures smoother 4D transitions and more coherent dynamic scene generation.

*(3) Controllable 4D Generation& Efficiency:* To evaluate the controllability and efficiency of our framework, we match the scene graph settings of LiDARCrafter for a fair comparison on the nuScenes dataset, denoted as Ours-sg. As demonstrated in Table 4, our scaffold-based framework with com-

pact NCR achieves superior 4D modeling results through a smaller representation. Furthermore, due to this compact representation, our method maintains high inference efficiency without additional computational burden, even when incorporating multi-stage progressive recoupling.

*Table 4.* Performance and Average Inference Time per Frame

| Method | Venue | Representation | JSD | MMD | Time (S) |
|---|---|---|---|---|---|
| LiDARGen | ECCV22 | Range | 5.74 | 2.39 | 67 |
| LiDMs | CVPR24 | Range Image | 5.86 | 0.73 | 3.5 |
| Uniscene | CVPR25 | Voxel | 31.55 | 13.61 | 2.1 |
| LiDARCrafter | AAAI26 | Range | 3.11 | 0.42 | 3.6 |
| Ours-sg | *present* | Scaffold | **2.87** | **0.39** | 3.1 |

*Table 5.* Ablation Study on Representation Settting

| Scaffold | NF | Multi-Resolution | Joint-Tuning | Point Cloud CD (↓) | F-Score (↑) |
|---|---|---|---|---|---|
| ✓ | ✓ | ✗ | ✓ | 0.3587 | 0.8401 |
| ✓ | ✓ | ✓ | ✗ | 0.3396 | 0.8493 |
| ✗ | ✓ | ✓ | ✓ | 0.3879 | 0.8257 |
| ✓ | ✗ | ✓ | ✓ | 0.3422 | 0.8473 |
| ✓ | ✓ | ✓ | ✓ | **0.3214** | **0.8521** |

### 4.2. Ablation Study

We conduct ablation studies to validate the effectiveness of key components in our method. *(1) Scaffold Setting.* As shown in Tab. 5, we ablate (i) Scaffold vs other Hexplane Representation (ii) Transformer NF vs other U-net NF, (iii) the *node multi-resolution feature* in the scaffold, and (iV) *joint training* $\mathcal{L}_{\phi_\alpha,\phi_\gamma}^{tuning}$ between the initial scaffold representation and the compact contourlet transformation stage. The multi-resolution node features provide fine-grained hierarchical encoding that improves geometric fidelity, while joint-tuning enables consistent optimization across both spatial and frequency domains. Removing either component leads to noticeable performance drops in both CD and F-Score. Specifically, enabling multi-resolution features improves Chamfer Distance by 10.4%, while joint-tuning further increases the F-Score from 0.8401 to 0.8521. Notably, the Scaffold representation plays a more essential role in maintaining point cloud fidelity than other modules, as its removal leads to the most significant performance degradation. These results demonstrate the complementary relationship between these components and the overall effectiveness of the proposed representation learning strategy.

*Table 6.* Ablation Study on NCR

| Filter | DB | Coarse2Fine | FRD (↓) | $MMD_{BEV}$ (↓) |
|---|---|---|---|---|
| Haar | ✓ | ✓ | 1728.2 | $8.83 \times 10^{-4}$ |
| Biorthogonal3-3 | ✓ | ✓ | 1421.4 | $2.51 \times 10^{-4}$ |
| Biorthogonal6-8 | ✓ | ✓ | 1205.3 | $1.54 \times 10^{-4}$ |
| Neural | ✓ | ✗ | 1126.7 | $1.14 \times 10^{-4}$ |
| Neural | ✗ | ✓ | 1143.5 | $1.28 \times 10^{-4}$ |
| Neural | ✓ | ✓ | **1087.4** | **$9.35 \times 10^{-5}$** |

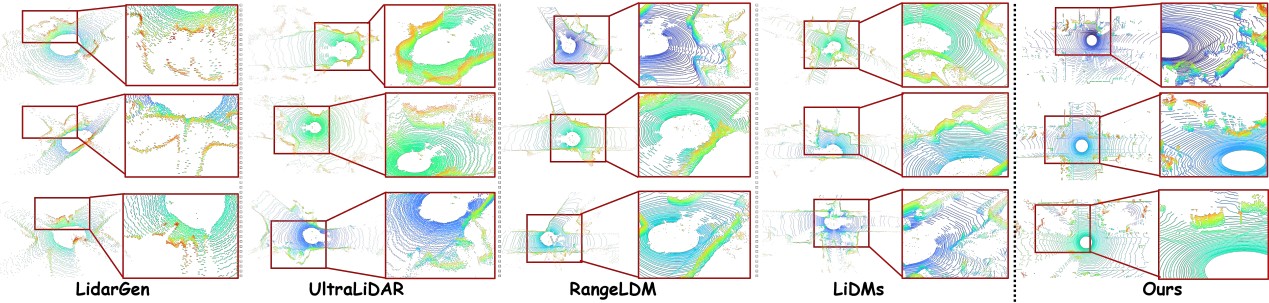

| LidarGen | UltraLiDAR | RangeLDM | LiDMs | Ours |

*Figure 3.* Visualization of lidar scene generation results against baselines.

*(2) NCR:* The filter design in the contourlet transform plays a crucial role in capturing data structure and spatial–frequency correlations. As shown in Tab. 6, we compare our learnable neural filter with several fixed filter types, including *Haar* and *Biorthogonal*. Our neural filter achieves significantly better results, improving FRD by 9.8% and F-Score by 39.3% compared with the best fixed Biorthogonal filter, demonstrating its superior adaptability to LiDAR structural characteristics. Moreover, since our framework performs explicit frequency decomposition, we further evaluate the directional bank and coarse2fine generation strategy. Enabling this strategy yields an additional 3.5% reduction in FRD and a 17.5% gain in F-Score compared with using the neural filter alone, showing that coarse2fine refinement enhances high-frequency detail reconstruction and DB gives explicit directional guidance improves the overall generative fidelity

*Table 7.* Ablation Study on Generation Scheme.

| DB | Recoupling | FRD ($\downarrow$) | MMD$_{BEV}$ ($\downarrow$) | JSD$_{BEV}$ ($\downarrow$) |
|---|---|---|---|---|
| ✓ | S&D | 1129.4 | $1.41 \times 10^{-4}$ | 0.095 |
| ✓ | S2D | 1105.2 | $1.13 \times 10^{-4}$ | 0.071 |
| ✗ | S2D | 1143.5 | $1.28 \times 10^{-4}$ | 0.087 |
| ✓ | S2D | **1087.4** | **$9.35 \times 10^{-5}$** | **0.059** |

*(3) Generation Scheme:* We further ablate the generation scheme by evaluating the effects of the *directional filter bank (DB)* and the *progressive re-coupling* strategy. Here, *S&D* denotes jointly generating static and dynamic components, while *S2D* represents generating dynamics conditioned on the static scene. As shown in Tab. 7, introducing the DB leads to improved reconstruction fidelity, as the directional positional embedding better guides recovery from coarse latent features. Moreover, the progressive re-coupling design enables spatial–temporal consistency between static and dynamic parts, yielding smoother and more coherent 4D generation. Together, these two components contribute to the best performance, reducing FRD and JSD while achieving the highest F-Score among all settings.

### 4.3. Application and Downstream tasks:

Thanks to the structural disentanglement in HieraScaffold, dynamic objects are naturally isolated from the static environment. We generate 10k data and labels for Sim10k by performing clustering on the dynamic point stream and assigning semantic categories (Vehicle/Pedestrian) via a boundingbox size heuristic. This allows for efficient, annotation-free generation of labeled training data. For LiDAR sampling, we strictly align the sensor viewpoints and scanning configurations with the native settings of the Waymo and KITTI datasets. We combine real data with our generated LiDAR samples to evaluate their effectiveness for model training. As presented in Tab. 8, the simulated data alone (*Sim10k*) already achieves performance comparable to real data, with only a 0.8 mIoU gap in segmentation and a 0.3 difference in detection accuracy. When augmenting with just 10% of real data (*Sim10k+1k*), our generated samples even outperform models trained solely on real data, demonstrating the diversity and generalization benefits of our synthetic LiDAR generation. See more visualization in suppl.

*Table 8.* Data Augmentation; Eval at Real data.

| Train Set | Segmentation (mIOU) | | | | Veh. Det. (IOU 0.7) | |
|---|---|---|---|---|---|---|
| | Overall | Vehcile | Peds. | B.g. | $\geq$1pt | $\geq$10pt |
| Real1k | 68.8 | 82.5 | 36.7 | 87.2 | 54.7 | 59.3 |
| Real10k | 73.9 | 87.0 | 41.9 | 92.8 | 60.0 | 65.9 |
| Sim10k | 73.1 | 86.3 | 41.4 | 91.5 | 59.7 | 65.5 |
| Sim10k+1k | **76.2** | **88.1** | **45.8** | **94.6** | **62.8** | **67.6** |

## 5. Conclusion

We introduced HieraScaffold, a unified hierarchical recoupling generation framework for scalable 4D LiDAR scene generation. HieraScaffold reconstructs continuous geometry from sparse scans via a dynamic unsigned distance scaffold and achieves compact, direction-aware modeling through a neural contourlet representation. The hierarchical re-coupling pipeline further enables coherent large-scale scene generation with coarse-to-fine refinement. Experiments on KITTI-360 and Waymo demonstrate that ours achieves superior geometric fidelity and temporal consistency over existing LiDAR generation methods.

## Impact Statement

The generation of large-scale outdoor scenes provides critical support for the rapid advancement of multi-task applications, including autonomous driving, virtual reality (VR), and augmented reality (AR). While these fields increasingly demand high-quality dynamic data, existing scene datasets suffer from severe limitations; traditional LiDAR sampling typically focuses on static environments, often overlooking the spatiotemporal coherence of the dynamic world and lacking essential geometric and temporal structural information.

Our framework addresses these challenges by introducing a high-fidelity 4D scaffold reconstructed from real-world samples, enabling a deeper understanding of dynamic environments. By leveraging the Compact Neural Contourlet Representation (NCR), we create an extremely efficient representation that facilitates 4D world comprehension while significantly expanding data modalities. This approach effectively circumvents the constraints of data scarcity and the inherent limitations of physical LiDAR sensors. Consequently, our proposed framework offers a unified, fully simulated solution for generating unbounded 4D worlds. This alleviates the need for manual map construction in simulators and overcomes the resource-intensive nature of large-scale data acquisition, providing a scalable foundation for future 4D perception and planning tasks.

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

# A. Framework Details

## A.1. Compact Neural Contourlet Representation

To achieve an efficient latent representation, we design a $K$-level Neural Contourlet Representation (NCR) compression scheme. This process is composed of two distinct stages: *Redundancy Minimization* for progressive spatial compression and *Core Disentangling* for high-frequency directional encoding. We provide a more detailed compact neural contourlet decompostion in Fig. 4(a).

**1. Redundancy Minimization:** In the first stage, we aim to progressively reduce the spatial dimensionality of the scaffold by removing spatiotemporal redundancy. For each level $l \in [0, K-1]$, we perform a Laplacian pyramid decomposition to downsample the input volume. Unlike the standard contourlet transform, we intentionally exclude the Directional Filter Bank (DFB) decomposition at these intermediate levels to focus solely on spatial compression. The learnable filters are optimized by minimizing the reconstruction loss, forcing the upsampled low-frequency core to approximate the input of the previous level with maximal fidelity:

$$\mathcal{L}_{recon} = \sum_{l=0}^{K-1} \|\text{Up}(\mathcal{C}_{l+1}) - \mathcal{C}_l\|_2^2 \qquad (11)$$

where $\mathcal{C}_l$ denotes the core representation at level $l$, and $\text{Up}(\cdot)$ represents the upsampling operation.

**2. Core Disentangling:** At the final level $K$, we perform the *Core Disentangling* to separate the fundamental structure from detailed variations. The compacted core $\mathcal{C}_K$ is decomposed into a coarse low-frequency component ($\mathcal{C}_{final}$) and a fine high-frequency component ($\mathcal{F}_{final}$). To efficiently capture the geometric details, the fine component $\mathcal{F}_{final}$ undergoes Directional Filter Bank (DFB) decomposition. Crucially, we apply *critical sampling* to map these directional sub-bands back to the original tensor dimensions of the core, ensuring a compact representation without coefficient expansion. This yields our final compact latent tuple ($\mathcal{C}_{final}, \mathcal{F}_{final}$), where $\mathcal{F}_{final}$ encodes the dense directional textures essential for high-fidelity generation.

**Implementation Details of NCR.** Specifically, the Directional Filter Bank (DFB) is implemented using learnable 3D convolutions with a kernel size of $5 \times 5 \times 5$. We set the number of directional bands to $K_j = 8$, which partitions the high-frequency spectrum into eight wedge-shaped directional sub-bands to effectively capture geometric singularities (e.g., edges and surfaces) along different orientations. Regarding the handling of 4D spatial-temporal volumes, we treat the temporal dimension $T$ as a batch dimension and apply the 3D DFB spatially to each frame. To ensure compactness, we adopt a *critical sampling* scheme (Higaki et al., 2008) that maps these $K_j$ directional sub-bands back to the

original tensor dimensions via frequency-domain checkerboard partitioning, thereby strictly preserving the coefficient count without redundancy expansion.

---

**Algorithm 1** Hierarchical Neural Contourlet Compression

---

**Input:** Initial Scaffold feature volume $\mathcal{S}$; Compression levels $K$; Learnable Neural Filters $\phi_\alpha, \phi_\gamma$.
**Output:** Compact Latent Representations $\mathcal{C}_{final}, \mathcal{F}_{final}$.
1: $\mathcal{C}_0 \leftarrow \mathcal{S}$ {Initialize core with input scaffold}
2: **// Stage 1: Redundancy Minimization (Spatial Compression)**
3: **for** $l = 0$ **to** $K-1$ **do**
4:     $\mathcal{C}_{l+1} \leftarrow$ Downsample($\mathcal{C}_l; \phi_\alpha$) {Laplacian pyramid decomposition}
5:     $\hat{\mathcal{C}}_l \leftarrow$ Upsample($\mathcal{C}_{l+1}; \phi_\gamma$) {Reconstruction for self-supervision}
6:     **Optimization Objective:** $\min_\phi \|\hat{\mathcal{C}}_l - \mathcal{C}_l\|_2^2$
7:     *Note: Directional Filter Bank (DFB) is excluded in this stage.*
8: **end for**
9: **// Stage 2: Core Disentangling (Directional Encoding)**
10: $\mathcal{C} \leftarrow \mathcal{C}_K$ {The final compact core cubes from last layer of Stage 1}
11: $\mathcal{C}_{final}, \mathcal{H}_{high} \leftarrow$ ContourletSplit($\mathcal{C}$) {Separate low/high frequencies}
12: $\mathcal{D}_{bands} \leftarrow$ DFB($\mathcal{H}_{high}$) {Apply Directional Filter Bank}
13: $\mathcal{F}_{final} \leftarrow$ CriticalSample($\mathcal{D}_{bands}$) {Map back to original tensor dimensions with position embedding}
14: **return** $\mathcal{C}_{final}, \mathcal{F}_{final}$

---

## A.2. Module Details of HRG:

**Block Chunk Generation:** Here, we avoid further subdivision along $Z$ for computational efficiency as well. Taking rightward generation as an example, the block-level generation can be represented as:

$$G_\rightarrow \left\{ \begin{array}{cc} \mathcal{D}_{[a]} & \mathcal{D}_{[b]} \\ \mathcal{D}_{[c]} & \underline{\mathcal{D}_{[d]}} \end{array} \right\} \xrightarrow{\text{Right}} \left\{ \begin{array}{cc} \mathcal{D}_{[b]} & \mathcal{D}_{[e]} \\ \underline{\mathcal{D}_{[d]}} & \mathcal{D}_{[f]} \end{array} \right\} \quad (12)$$

where $\mathcal{D}[a\text{–}d]$ denote the currently generated spatio-temporal blocks along the two horizontal axes, $\underline{\ldots}$ is the consistent part, and $\mathcal{D}[b, e; d, f]$ represent the adjacent blocks to be generated in the rightward direction conditioned on the existing scaffold context.

**Generation stream from Guassian to 4D scene.** We give a more detailed illustration in Fig. 4(b), it starts from a Gaussian noise to 1) generate the static coarse, 2) refine static fine from coarse with directional bank position embedding. We then recover the complete static scaffold from these two compoment using reverse NCR process, and then to

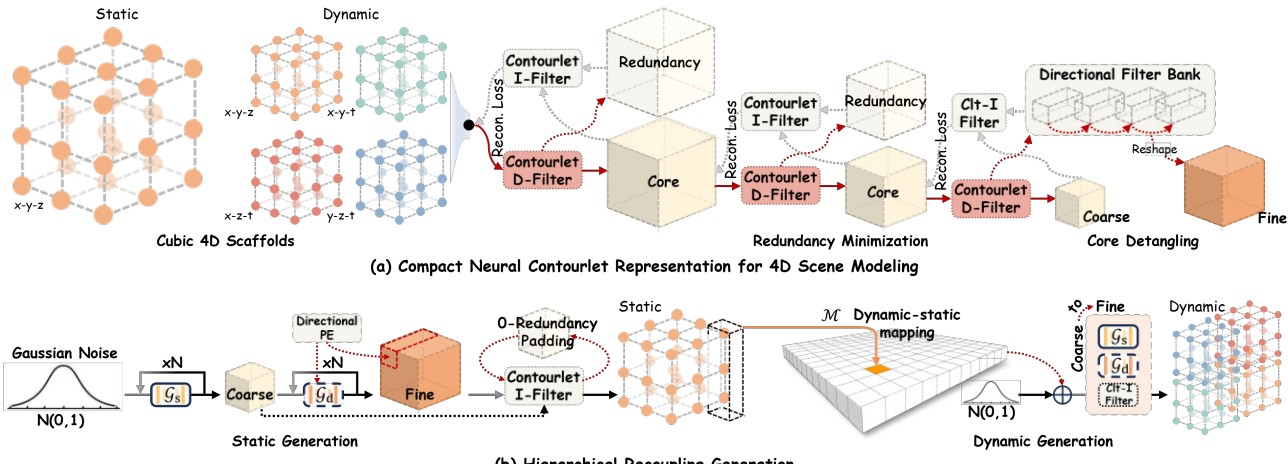

*Figure 4.* (a) Compact neural contourlet representation construction. (b) Framework of the proposed HRG flow.

recouple static into dynamic process with a dynamic-satic mapping and repeat the coarse2fine for dynamic scaffold. In the end, two scaffolds can be assembled to the final 4D scenes.

### A.3. More Visualizations:

We demonstrate the multiple block chunk assembled large scale of 4D scene in Fig. 5, which support various tasks in 4D world including lidar sample, dense point geometry acquirement, semantic segmentation and object detection. Fig. 6 further shows large-scale generation results.

### A.4. Application and Downstream tasks:

Fig. 7 shows the results of our inpainting application, enabling flexible scene re-editing. We also demonstrate controllable conditional generation, thanks to our proposed framework, fine-tuned with ControlNet, the model allows intuitive control over the generated results. As shown in Fig. 8, text-based conditioning enables generating scenes such as a *busy urban road* and sampling LiDAR point clouds across multiple time steps. Moreover, our framework can easily produce large-scale simulated data to support downstream tasks such as semantic segmentation and object detection (Fig.1).

## B. Why is Scaffold

**Reason for Naming:** The term Scaffold is motivated by how our representation is constructed around the raw LiDAR points. Since LiDAR only samples sparse measurements near the underlying surfaces, we build our representation by placing learnable nodes in the local neighbourhood of these points to infer the zero-level set of the surface. Regions far from the observed samples do not require dense fitting. This process resembles assembling a structural scaffold around a

building during construction, which supports and outlines the shape without filling the entire volume. Hence the name Scaffold representation.

**Computational Advantage:** A second advantage comes from the computational behaviour of this design. Because the Scaffold is formed only around surface-adjacent regions, the complexity during fitting depends mainly on the geometry of the surface instead of the full 3D voxel volume. This allows the representation to scale according to surface extent rather than cubic spatial resolution, making the basic geometric modeling far more efficient for LiDAR data.

**Help for redundancy minimization:** The structured 3D layout of the Scaffold is beneficial for later compact modeling. When converting the Scaffold into a frequency-based compact form, unobserved spatial regions are filled with fixed-form placeholder nodes. This explicit and well-defined redundancy makes our redundancy minimization stage more stable and more effective, providing support for the overall design of our compact representation pipeline.

**Why Scaffold helps 4D:** Scaffold directly centers the representation on the true geometric carriers of 4D change, the surfaces sampled by LiDAR, rather than the full spatial volume. By aligning learnable nodes to these surface regions, Scaffold preserves stable geometric anchors across time, making temporal correspondence explicit instead of implicitly inferred through dense voxel updates or image-based projections. This surface-aligned structure allows dynamic motion to be modeled where it actually occurs, while static regions remain consistently grounded, preventing drift and temporal fragmentation. As a result, both spatial structure and temporal evolution are encoded in a way that is computationally efficient and naturally coherent. This gives Scaffold a decisive advantage for large-scale 4D LiDAR generation, where maintaining consistent geometry over long horizons is essential.

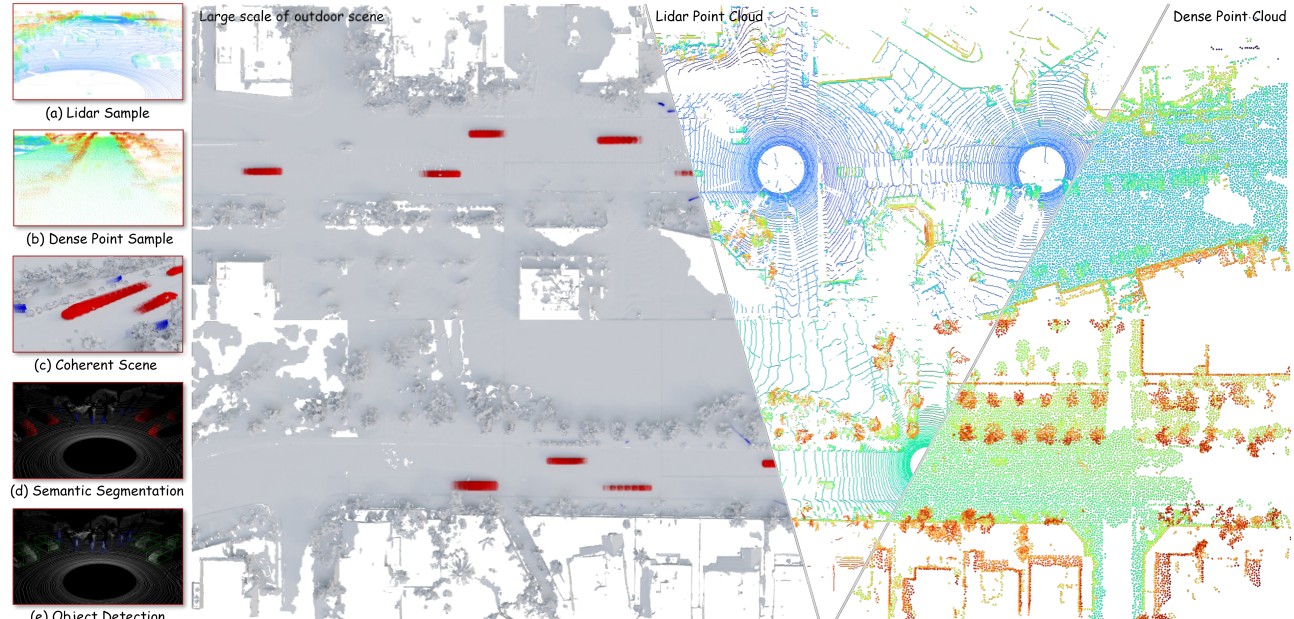

*Figure 5.* We propose HieraScaffold to generate dynamic LiDAR scenes with fine-grained geometry and realistic temporal coherence, producing grounded outdoor scenes, sampled LiDAR, and dense point clouds for large-scale modeling (a-c) and downstream tasks (d,e).

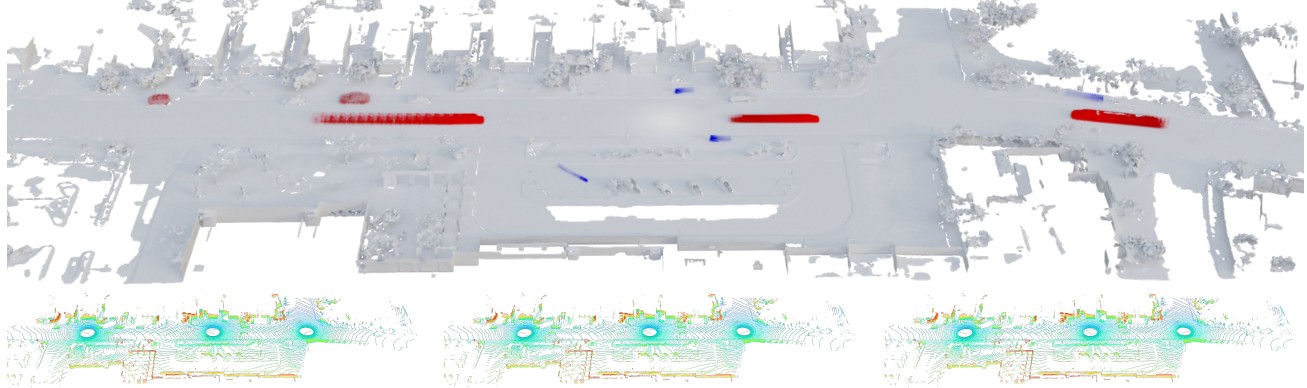

*Figure 6.* Visualization for large scale of scene generation.

## C. Limitation and Discussion

**Limitation and Discussion.** Our framework is trained on existing LiDAR datasets, which currently form the main constraint on scene diversity. In principle, the model can represent any coherent dynamic environment, indoor or outdoor, provided that raw dynamic point clouds are available. However, real LiDAR scans often contain large unobserved regions where no returns are recorded. These areas are not truly empty but simply unscanned, and the model may reproduce such gaps during generation, leading to spatial holes that do not reflect the actual scene structure.

As highlighted in the red-box regions of Fig. 9, the boundary discontinuities are caused by incomplete LiDAR observations rather than true geometric structure. Since these areas were never scanned in the raw dataset, the model has no valid surface cues to follow and therefore terminates generation along those directions. This behavior reflects the

data limitation rather than a failure of the generation process itself, and improving coverage in future dynamic LiDAR datasets will be essential for resolving such boundary stops.

**Future work.** In future work, we plan to construct a dedicated dynamic large-scale LiDAR dataset with richer motion patterns and more complete spatial coverage across various object categories. This will further strengthen the model's ability to learn stable dynamic geometry. In addition, improving robustness to incomplete observations and enhancing the model's capability to recover missing structures will be an important direction for future development.

## D. Additional Visual Results

We present additional qualitative comparisons of our generated LiDAR scenes against recent baselines. Since LidarDM shows notably weaker performance compared with mainstream diffusion based methods such as RangeLDM and

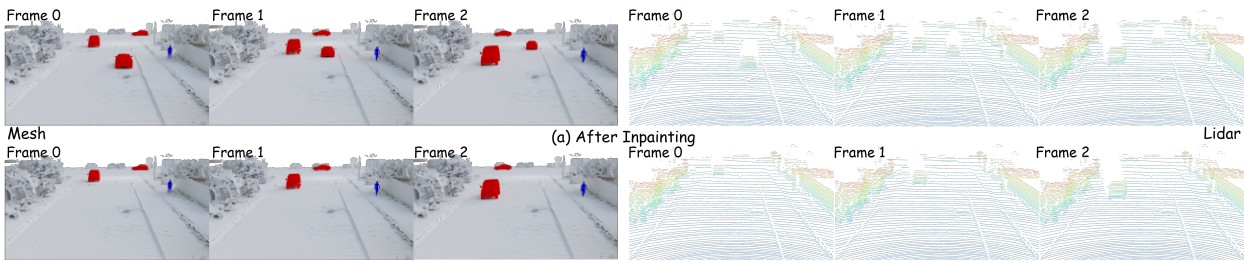

Figure 7. Visualization of large scene generation.

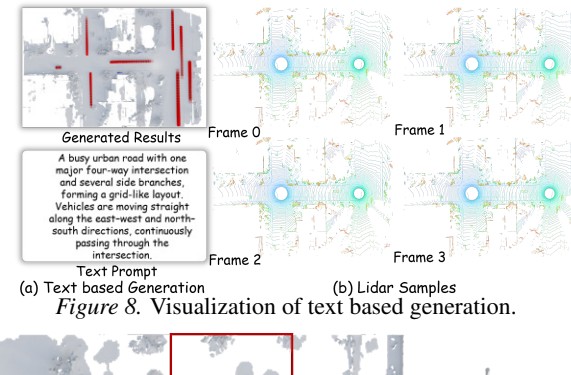

*Figure 8.* Visualization of text based generation.

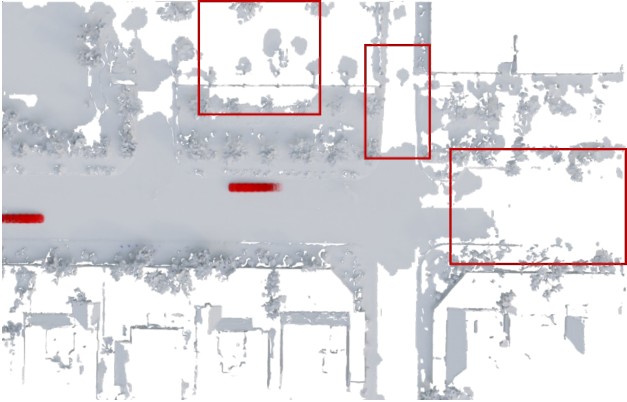

*Figure 9.* Dataset limitation demonstration and failure region.

LiDMs, we focus our discussion on the more representative approaches. As shown in Fig. 3, LidarGen produces limited global structure, with scene geometry insufficiently unfolded. Other diffusion-based methods exhibit stronger structural modeling, but still suffer from clear degradation in long-range geometry, where distant regions become blurry or collapse into unstable shapes.

In contrast, our method maintains stable structural fidelity across both near and far distances. By leveraging the scaffold representation to encode spatial and temporal relationships, our generation is not affected by distance-related degradation. As a result, our model preserves strong geometric cues, including vehicle shapes, wall structures, and other latent patterns present in the training scans, even when such structures are not explicitly annotated. This yields more realistic and coherent LiDAR scenes compared with existing methods.

Moreover, We also provide an animated visualization in GIF format, included in the supplementary files, to further illustrate the dynamic samples of our generated 4D LiDAR scenes.

## E. Details of Experimental Setting

**Datasets:** We train and evaluate our method on two outdoor LiDAR datasets: KITTI-360 dataset (Behley et al., 2019; Geiger et al., 2013) and Waymo Open Dataset (Sun et al., 2020). KITTI-360 serves as the standard benchmark for evaluating lidar generation and reconstruction tasks. The dataset consists of nine driving sequences containing a total of 76,715 lidar frames captured across Karlsruhe, Germany, using a 64-beam Velodyne HDL-64E sensor. Following prior work, we adopt sequence 0 (11,518 samples) for validation and the remaining eight sequences (65,197 samples) for training. Waymo Open Dataset provides a large-scale collection of lidar sequences captured from diverse urban and suburban environments across multiple cities in the United States. The dataset contains 798 training sequences and 202 validation sequences.

**Baselines:** We compare our method against a comprehensive set of state-of-the-art lidar reconstruction and generation approaches. For the reconstruction evaluation, we include NSKR (Huang et al., 2023a), lidar-DM (Zyrianov et al., 2025), and K-Planes (Fridovich-Keil et al., 2023). These methods represent different levels of spatial–temporal modeling capacity: NS-KR focuses on static *3D-only* reconstruction, lidar-DM introduces partial dynamic modeling, while K-Planes extends to full *4D* representation learning. This progressive comparison allows us to evaluate the reconstruction fidelity and representation efficiency of our proposed hierarchical scaffold framework under varying degrees of spatio-temporal complexity. For the generation evaluation, we benchmark our method against state-of-the-art diffusion- and transformer-based lidar generative models, including LiDMs (Ran et al., 2024), Ultralidar (Xiong et al., 2023), RangeLDM (Hu et al., 2024a), and lidar-DM (Zyrianov et al., 2025). All generative baselines are trained or fine-tuned under identical data splits and preprocessing pipelines to ensure fair comparison. We report both frame-wise reconstruction metrics and 4D sequence-level genera-

tion consistency to comprehensively assess the performance of our framework.

**Evaluation Metrics:** We evaluate reconstruction quality using Chamfer Distance and F-score (5 cm). For generative performance, we report FRD, MMD-BEV, and JSD-BEV to assess geometric accuracy, structural consistency, and distributional diversity of the synthesized 4D LiDAR scenes. We evaluate reconstruction quality on both lidar point clouds and their corresponding range images using Chamfer Distance (CD) and F-score with a threshold of 5 cm. To further assess the generative quality and diversity of the synthesized 4D lidar scenes, we additionally report Fréchet Range Distance (FRD), Mean Minimum Distance in BEV ($MMD_{BEV}$), and Jensen–Shannon Divergence in BEV ($JSD_{BEV}$). These complementary metrics jointly measure geometric accuracy, structural consistency, and distributional diversity across generated and real lidar samples.

**Stop condition for block–chunk generation.** Our $2 \times 2$ block–chunk generation strategy ensures local spatial and temporal consistency during expansion. However, if applied uniformly across all horizontal directions, it may lead to over-extended dense regions, producing scenes that resemble compact urban blocks and reducing the diversity of large-scale layouts. To avoid this spatial flattening effect, we introduce a stop condition based on the unsigned distance ratio along the boundary interface. Specifically, when the proportion of near-surface UDF values on a boundary falls below 10% in a given horizontal direction, we halt further expansion along that direction and instead continue generation in alternative directions. Combined with random rotation augmentation during training, this adaptive stopping mechanism preserves spatial diversity and produces scene layouts that align more naturally with real-world outdoor distributions.

*Table 9.* Human evaluations: 100 generated LiDARs rated on a scale of 1 to 5 by 30 people.

| Human rate (↑) | 3D Consistency (↑) | Artifacts (↓) | Detail Score (↑) |
|---|---|---|---|
| LiDARGen | 2.5 | 4.6 | 1.9 |
| UltraLiDAR | 2.8 | 4.2 | 3.1 |
| LiDMs | 3.9 | 3.2 | 4.3 |
| RangeLDM | 3.7 | 3.5 | 4.2 |
| LidarDM | 3.1 | 4.0 | 3.9 |
| **Ours** | **4.3** | **1.3** | **4.8** |

## F. Human Evaluation

We conduct a perceptual user study to assess the quality of our generated LiDAR scenes compared with the baselines. A total of 30 participants rated 100 randomly sampled generated LiDAR frames from each method. The evaluation consists of three criteria:

- **3D Consistency** (↑): measures how well the global spatial structure aligns across the scene, reflecting coherent geometric layout.

- **Artifacts** (↓): evaluates the presence of unrealistic patterns such as shape collapse, missing walls, or distorted surfaces.

- **Detail Score** (↑): rates the fidelity of fine-grained structures, such as the shape of cars, poles, pedestrians, and other small objects.

Participants were asked the following three questions for each sample:

1. **How consistent is the 3D structure of the scene?** (1 = inconsistent, 5 = highly coherent)

2. **How many unrealistic artifacts do you notice?** (5 = many artifacts, 1 = almost none)

3. **How clear and detailed are the detailed structures?** (1 = poor details, 5 = highly detailed)

As shown in Tab. 10, we conduct a human evaluation on 100 LiDAR samples generated by our method and all baselines, rated on a 1–5 scale by 30 participants. Our approach achieves the highest scores across all criteria: it delivers the best 3D structural consistency (4.3), the fewest artifacts (1.3, lower is better), and the most detailed geometric structures (4.8). These results indicate that our generations are perceived as more coherent, cleaner, and richer in fine-grained details compared with existing methods.

## G. More results

**Scaffold Hierarchy**: Implemented via $L = 3$ levels of learnable multi-resolution feature grids at each node. As Table 5 demonstrates, removing these node features degrades Chamfer Distance by 11.6% (0.3214 to 0.3587), proving this structure is crucial for capturing fine-grained geometric fidelity. Additional results varying hierarchy levels while keeping the number of features constant are presented below:

*Table 10.* Scaffold hierarchy: performance under set levels of Hierarchy.

| Hierarchy Level | 1 | 2 | 3 | 4 |
|---|---|---|---|---|
| CD | 0.3587 | 0.3342 | **0.3214** | 0.3367 |
| F-Score | 0.8401 | 0.8507 | **0.8521** | 0.8501 |

**Learning from GT-free data**: P1. Label-Free Core Pipeline. Our framework encodes and generates all foreground instances uniformly within the continuous spatial-temporal scaffold decomposition. As shown in the top red

box of Fig. 10(a), this process is entirely class-agnostic and requires no semantic labels. During inference, our model predicts pure geometry, not semantic classes. Similar to 2D K-Planes (CVPR23), it naturally learns the static-dynamic separation process without additional annotations. The yellow region represents an optional annotation-guided enhancement step to complete occluded dynamic geometry.

P2. GT-Free Framework for Scalable Deployment. Figure Hiera4D(b) details our pipeline for GT-free data. Our key insight is that off-the-shelf segmentation and point cloud preprocessing methods can achieve the necessary geometric enhancement without GT:

A. We first apply the ERASOR (Lim et al., 2021) algorithm to separate the dynamic foreground from the static background.

B. The static background is processed by our neural field as usual. For the unlabeled dynamic points, we adopt a cluster-then-complete strategy.

C. As shown in the red box, we use DBSCAN to isolate most dynamic instances, exploiting the natural safe driving distances present in outdoor scenes. We then assign categories using heuristic bounding box size checks (which easily distinguish between cars, pedestrians, and cyclists). For hard samples, such as closely adjacent pedestrians (visualized in Fig. 10(d)), we apply a finer segmentation model (Park et al., 2019) and repeat the size check. Only successfully categorized instances are sent to Point-Tr for completion; unrecognized clusters bypass this step.

D. Finally, all dynamic instances (both enhanced and by-passed) are combined for self-supervised dynamic UDF learning. As emphasized in point 1, 4D scaffold encoding requires only spatial-temporal geometric UDFs, completely bypassing the need for semantic labels.

We greatly appreciate your insightful point regarding the necessity of a clear GT-free pipeline, which is indeed crucial for real-world deployment and wider adoption. While instance isolation serves primarily as a modular preprocessing step in our framework, allowing our core designs (the 4D Scaffold, NCR, and HRG generation in the red box of Fig. 10(a)) to remain purely geometry-driven, we fully agree that making this entire pipeline completely GT-free is practically significant. To ensure our 4D framework is fully scalable to unlabeled data, we will thoroughly document and release all corresponding preprocessing scripts and clustering details alongside our core codebase.

**Mathematical Analysis of Compression Efficiency**: Our motivation to decompose the 4D space into four 3D sub-spaces aims to find the optimal balance between unambiguous geometric modeling and computational efficiency:

A. The Ambiguity of 2D Planes: Projecting 4D dynamics directly onto 2D planes (with a complexity of $6N^2$, like K-Planes) crosses too many dimensions, causing severe feature entanglement and structural overlap for inherently sparse LiDAR data. While sufficient for volume rendering in NeRFs, these highly entangled 2D projections fail to provide the unambiguous geometric priors strictly required by generative diffusion models.

B. The Intractability of 4D Voxels: Conversely, a direct 4D voxel grid ($\mathcal{O}(N^4)$) is computationally prohibitive. Deploying heavy generative networks (e.g., 3D/4D DiT) on dense 4D tensors for large-scale autonomous driving scenes leads to instant memory overflow and intractable training times.

C. The Superior Efficiency of 3D Subspaces + NCR: Our design reduces the initial complexity from $\mathcal{O}(N^4)$ to $4N^3$, explicitly avoiding the catastrophic projection loss of 2D planes. More importantly, our Neural Contourlet Representation (NCR) explicitly extracts directional information to maximize redundancy removal with minimal information loss. Under a 3-level NCR compression, our generative latent complexity drastically shrinks from $4N^3$ to $4(N/2^3)^3 = \frac{1}{128}N^3$.

Mathematically, $\frac{1}{128}N^3 < 6N^2$ holds strictly true for any resolution $N < 768$. Given that typical latent resolutions $N$ for diffusion models are 128 or 256, our compressed 3D representation is factually more computationally efficient than K-Planes' 2D approach. This allows us to encode significantly higher spatio-temporal resolutions under an identical parameter budget.

**NCR Compression Analysis**:Assuming a latent feature grid size of $256 \times 256 \times 32 \times 32$, we compare our NCR module against a 2D K-Planes baseline. We use $a/b$-NCR to denote $a$ static and $b$ dynamic levels of multi-scale NCR compression:

*Table 11.* Ablation Study of NCR.

| Version | Params | CD | F-Score | inference time (s) |
|---------|--------|------|---------|--------------------|
| K-Planes | 99.3k | 0.1244 | 0.9040 | 3.87 |
| No-NCR | 4.7M | 0.0925 | 0.9241 | OOM |
| 1/1-NCR | 589.8k | 0.0931 | 0.9236 | OOM |
| 2/2-NCR | 73.7k | 0.0934 | 0.9229 | 3.43 |
| 2/3-NCR | 37.8k | 0.0946 | 0.9214 | 2.3 |

Massive Redundancy Minimization: The "No-NCR" baseline demonstrates that while a dense 4D representation achieves the best metric, it requires a massive 4.7M parameters just for the feature grid, making generative modeling computationally intractable. By applying frequency-domain compression (e.g., 2/2-NCR), we dramatically shrink the representation size down to 73.7k parameters with a negligible drop in reconstruction quality, significantly easing the computational burden of the generation phase. Moreover, our decoupled Scaffold allows static and dynamic elements to be compressed independently; even when static redun-

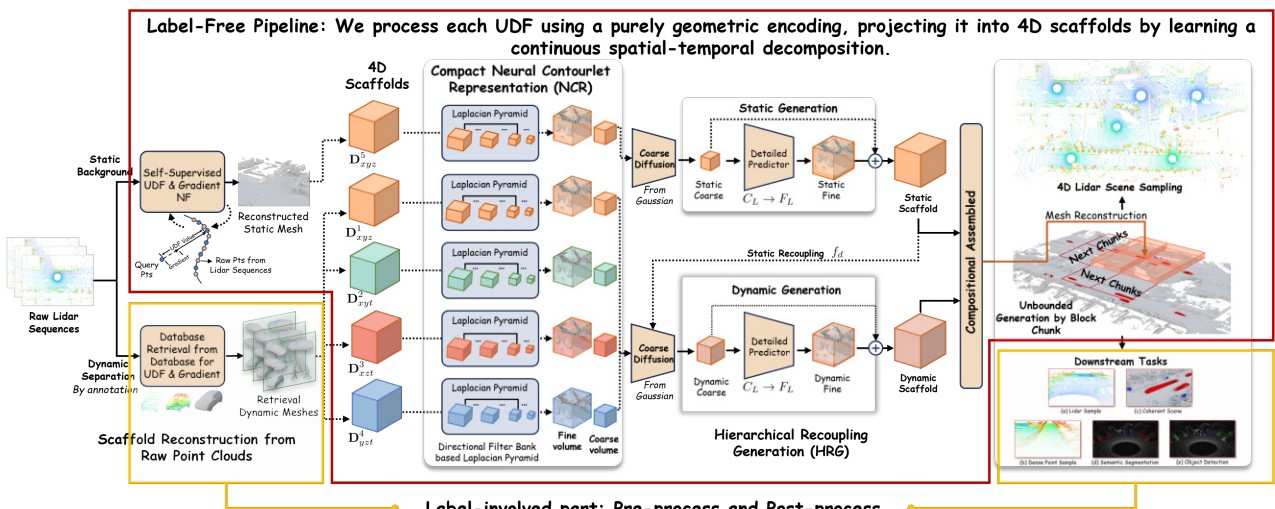

(a) Label Illustration of the Label-Free and Label-Guided pipelines. The purely geometric encoding naturally handles unknown dynamics, while the label-involved module serves pre-process and post-process for data enhancement and downstream tasks.

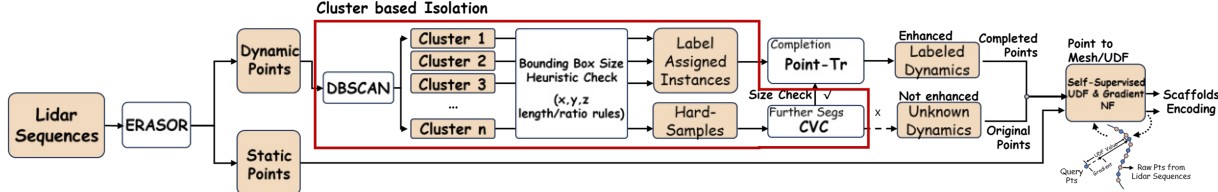

(b) GT-Free Framework: Isolated instances assigned a category via heuristics undergo point cloud completion, while unknown dynamics bypass this step to remain as raw partial scans. Both streams are subsequently unified for self-supervised UDF reconstruction prior to 4D Scaffold Encoding.

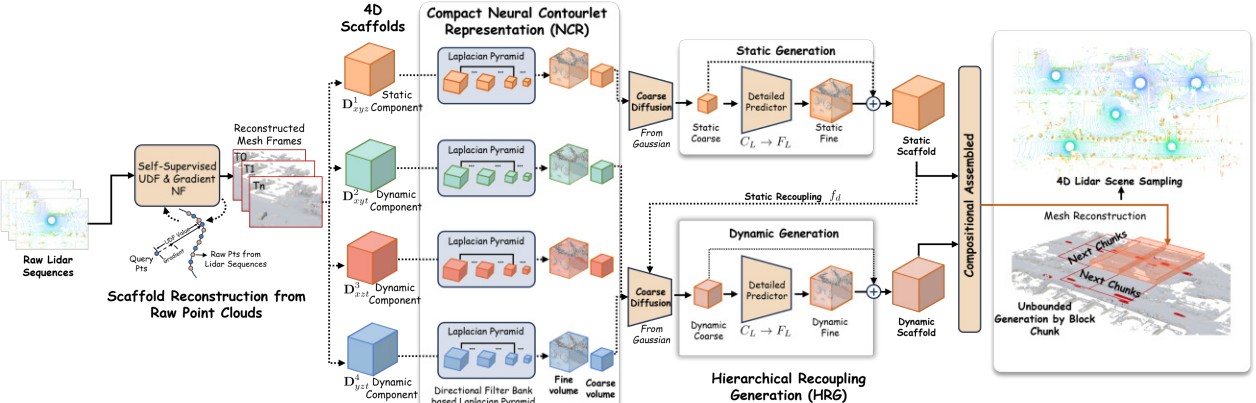

(c) No static-dynamic separation pipeline without annotation. All UDFs are encoded into the same 4D scaffolds.

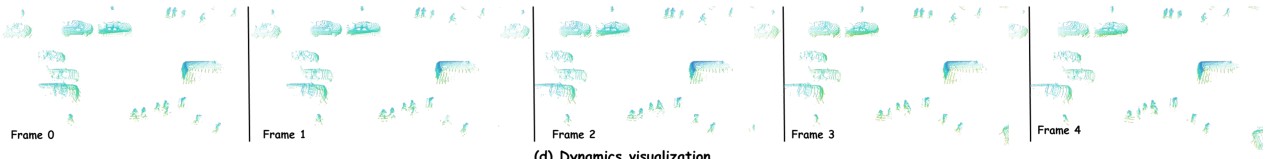

(d) Dynamics visualization

*Figure 10.* GT-free Learning Framework Illustration.

dancy is minimized to its limit, we can further compress the dynamic components (2/3-NCR) to achieve faster inference speeds with negligible loss in accuracy.

