# OpenReview forum: "HieraScaffold: Learning Compact Hierarchical Representations for Scalable 4D LiDAR Generation"
_ICML.cc/2026/Conference — ICML 2026 regular_

### Official Review · Reviewer_523G · 2026-03-04

**Soundness:** 4
**Presentation:** 4
**Significance:** 3
**Originality:** 3
**Overall Recommendation:** 4
**Confidence:** 4

**Summary:**

This paper focuses on 4D LiDAR point cloud video generation in autonomous driving scenarios, addressing key issues of traditional methods. HieraScaffold integrates static-dynamic decoupling, Scaffold representation, NCR frequency-domain compression, and S2D conditional generation. It splits scenes into static background and dynamic objects, which are modeled by separate networks, and then fuses features to output continuous, temporally smooth 4D point cloud sequences. Experiments on KITTI-360 and Waymo datasets verify its effectiveness in reconstruction accuracy, generation quality, and temporal consistency.

**Compliance With Llm Reviewing Policy:**

Affirmed.

**Final Justification:**

The paper has key strengths that align with initial positive evaluations. The authors’ rebuttal effectively addresses most initial weaknesses. Key evaluation dimensions remain strong, with result unchanged.

**Key Questions For Authors:**

1.Dynamic generation highly depends on static reconstruction, which may lead to error accumulation and propagation. It is worth discussing whether robustness constraints or fault-tolerant mechanisms can be introduced to alleviate such issues.

2.A smoother fusion strategy or explicit spatial mask could be explored to improve geometric consistency, rather than taking the minimum of UDFs.

3.Strategies such as static-dynamic decoupling and S2D generation have been widely used. It would be helpful if the authors could more clarify the unique contributions of this work in representation and modeling to highlight its novelty.

4.The influence of NCR frequency-domain module on model efficiency, parameters, and computation complexity has not been systematically analyzed. Quantitative results would help clarify the actual gains brought by the frequency-domain representation.

**Limitations:**

Although the proposed method achieves favorable performance in point cloud quality and temporal smoothness, it still has limitations: it relies on dataset annotations and external Mesh resources, the static background cannot be dynamically updated with ego-motion, and sufficient ablation studies and complexity analysis for frequency-domain compression are lacking.

**Strengths And Weaknesses:**

Strengths:

1.Adopting the static-dynamic decoupling modeling method, training static background and dynamic objects separately, effectively improving the temporal consistency of point cloud sequences, and fundamentally solving common unreasonable phenomena such as motion drift and dynamic object penetration in traditional methods.​

2.Introducing sparse scaffold representation to replace traditional voxel and point cloud representation methods, it is both lightweight and efficient, which can not only effectively store scene geometric features, but also significantly improve the training and inference speed of the model.​

3.Applying NCR neural contourlet transform to 4D LiDAR generation, its multi-scale and directional decomposition characteristics are suitable for the strong directional structure of LiDAR scenes, which can well retain scene detail information while compressing data redundancy.​

4.Proposing the S2D static condition-guided generation strategy, using the geometric features of the static background as constraints to guide the generation of dynamic objects, effectively ensuring the spatial consistency between dynamic objects and static scenes, and avoiding unreasonable situations such as floating.​

5.Experiments are carried out based on two mainstream autonomous driving datasets, KITTI-360 and Waymo, with multi-dimensional evaluation indicators designed. The experimental verification is adequate and the results are highly convincing.

Weaknesses:

1.Dynamic generation relies heavily on static reconstruction. Errors in the static background may lead to unreasonable dynamic generation results, so the model’s robustness and error correction need further analysis.

2.The static-dynamic fusion strategy uses UDF minimum without explicit boundary handling, which may causes artifacts and requires better geometric consistency.

3.The experimental design is insufficient: it lacks ablations for core modules  (Scaffold, NCR, static-dynamic decoupling), so their necessity is not fully validated.

4.The NCR module lacks analysis of model parameters and computational complexity.

5.The actual efficiency and performance impact of frequency-domain compression are not sufficiently quantified.

---

> ### Author Rebuttal · Authors · 2026-03-31
>
> W1: We've added a dedicated discussion on robustness and error correction mechanisms across three areas:
>
> A. Implicit Fault Tolerance: Our Dynamic-Static Mapping ($\mathcal{M}$, Eq. 10) uses soft cross-attention, enabling the dynamic generator to adaptively down-weight noisy static regions. This naturally prevents minor static artifacts from corrupting dynamic placement.
>
> B. Explicit Robustness: To mitigate error accumulation, we introduce Condition Perturbation. Injecting mild Gaussian noise or feature dropout into the static condition $\mathcal{G}_s(D^5)$ during training forces the dynamic diffusion model to learn generalized motion priors instead of overfitting to perfect reconstructions.
>
> C. Future Physical Constraints: Integrating explicit physical-check methods (e.g., GPLD3D, BrickGPT) to strictly enforce geometric validity and collision-free interactions is a highly promising direction for future work.
>
> W2: We agree that high-fidelity surface reconstruction requires explicit boundary handling. Our initial hard $\min(\cdot)$ choice was pragmatic, as sparse LiDAR is relatively insensitive to fine $C^0$ discontinuities at intersections.To alleviate boundary artifacts, we adopted a Polynomial Smooth Minimum. New tests confirm this smoother fusion effectively enhances local point cloud quality: the local Chamfer Distance (CD) in a 0.5m × 0.5m intersection area improved from 0.0535 to 0.0519. Range view metrics remain nearly identical, as sparse 2D projections are insensitive to these fine boundaries. We will add this quantitative ablation to formally demonstrate the improved geometric consistency.
>
> W3: Core ablations are already detailed in Sec. 4.2, which we will make more explicit. Scaffold necessity is evaluated in Table 5 ("No Scaffold" replacing it with a 2D HexPlane). NCR contribution is analyzed in Table 6; we will also add a strict "No NCR" baseline. Static-dynamic decoupling is studied in Table 7 (S2D ablation), and we have added a direct "No Decoupling" baseline yielding FRD: 1226.7, MMD: $1.54\times 10^{-4}$, and JSD: 0.177.
>
> W4&5: We have added a comprehensive ablation for parameter analysis and frequency-domain compression impacts. Assuming a latent feature grid size of $256 \times 256 \times 32 \times 32$, we compare our NCR module against a 2D K-Planes baseline. We use $a/b$-NCR to denote $a$ static and $b$ dynamic levels of multi-scale NCR compression:
> | Version | Params | CD | F-Score| inference time (s)|
> | -------- | -------- | -------- | -------- | -------- |
> | K-Planes    | 99.3k     | 0.1244     | 0.9040 | 3.87 |
> | No-NCR    | 4.7M     | 0.0925     | 0.9241 | OOM |
> | 1/1-NCR    | 589.8k     | 0.0931     | 0.9236 | OOM |
> | 2/2-NCR    | 73.7k     | 0.0934     | 0.9229 | 3.43|
> | 2/3-NCR    | 37.8k     | 0.0946     | 0.9214 | 2.3 |
>
> Massive Redundancy Minimization: The "No-NCR" baseline demonstrates that while a dense 4D representation achieves the best metric, it requires a massive 4.7M parameters just for the feature grid, making generative modeling computationally intractable. By applying frequency-domain compression (e.g., 2/2-NCR), we dramatically shrink the representation size down to 73.7k parameters with a negligible drop in reconstruction quality, significantly easing the computational burden of the generation phase. Moreover, our decoupled Scaffold allows static and dynamic elements to be compressed independently; even when static redundancy is minimized to its limit, we can further compress the dynamic components (2/3-NCR) to achieve faster inference speeds with negligible loss in accuracy.
>
> Q3-Unique Contributions: We sincerely thank the reviewer for this insightful comment. While static-dynamic decoupling is a widely used and proven paradigm in rendering-based novel view synthesis (e.g., DynamicNeRF-ICCV21, Spacetime Gaussian-CVPR24), its potential remains largely unexplored in the domain of 4D LiDAR generation. Our unique contributions in representation and modeling move beyond simply borrowing this concept; rather, we specifically address the structural-fidelity versus computational-cost trade-off inherent in generative modeling through two distinct innovations:
>
> A. Decoupled NCR for Extreme Efficiency: Separating spatiotemporal redundancy pruning allows tailored compression. Applying an extra NCR level to highly sparse dynamic parts (2/3-NCR) requires only ~38% of the K-Planes parameters (37.8k vs. 99.3k) while capturing significantly more accurate spatiotemporal structures (CD: 0.0946 vs. 0.1244; F-Score: 0.9214 vs. 0.9040).
>
> B. Soft Recoupling Mechanism: Instead of rigid, hard-coded geometric combinations, our soft mechanism learns underlying static-dynamic mapping patterns. This hierarchical recoupling successfully maintains an extremely compact representation while enabling high-precision reconstruction of complex 4D scenes.

---

> > ### Author Rebuttal · Reviewer_523G · 2026-04-02
> >
> > I appreciate the authors' detailed response, which has fully addressed my concerns. I will retain my original score.

---

> > > ### Author Response · Authors · 2026-04-02
> > >
> > > Thank you for your constructive comments and for reviewing our rebuttal. We are thrilled that your concerns have been fully resolved. Your valuable suggestions have greatly helped us improve the quality of our work. We deeply appreciate your time and support.

---

### Official Review · Reviewer_8t8k · 2026-03-09

**Soundness:** 3
**Presentation:** 3
**Significance:** 2
**Originality:** 2
**Overall Recommendation:** 4
**Confidence:** 4

**Summary:**

This paper proposes HieraScaffold, a hierarchical generative framework for 4D LiDAR scene synthesis. The approach (i) learns a multi-resolution scaffold neural field that predicts unsigned distance fields (UDF) and gradients from raw LiDAR sequences while factorizing static and dynamic components into volumetric bases; (ii) compresses these scaffolds via a learnable neural contourlet representation (NCR) using a Laplacian pyramid plus directional filter banks; and (iii) performs hierarchical recoupling generation with a coarse-to-fine pipeline that first generates the static scene and then conditions dynamic content on it, with a block-chunk strategy for unbounded scenes. On KITTI-360 and Waymo, the method reports improvements over range- and diffusion-based baselines on FRD, MMD-BEV, and JSD-BEV, and shows better temporal consistency metrics.

**Compliance With Llm Reviewing Policy:**

Affirmed.

**Final Justification:**

I appreciate the authors' rebuttal on the GT information. Now I understand that the framework can be annotation-free. I tend to raise my score to the positive side.

**Key Questions For Authors:**

What's your insight to use the ground-truth bounding boxes to distinguish moving object from the static background? How do you deal with the unknown moving objects?

**Limitations:**

yes.

**Strengths And Weaknesses:**

Strengths:
1. The proposed dynamic scaffold representation effectively transitions from sparse point clouds to continuous unsigned distance fields. This strategy provides a robust and mathematically sound geometric foundation for downstream generation.
2. The Neural Contourlet Representation (NCR) is a clever design choice. By transforming the dense 3D/4D spatial features into a frequency domain and applying critical sampling, the method achieves a compact latent space that significantly reduces the computational burden of 3D diffusion models.
3. The empirical validation is thorough and shows impressive results. As shown in Table 1, the reconstruction F-score on KITTI-360 reaches 0.9214, demonstrating the high fidelity of the scaffold representation.

Weaknesses:
1. The K-Planes create 2D planes for faster computation by using mature 2D operators such as convolution. In this paper, the 4D space are decomposed to four 3D subspaces. However, the computation on 3D voxels are not highly optimized compared with direct 4D voxels. I cannot get the motivation here.
2. The introduction asserts that the framework models 4D structures "directly from raw LiDAR sequences". However, Section 3.1 explicitly details that the scene is decomposed "by labelled annotations" and that dynamic object modeling relies on querying "object meshes from a pre-defined database to directly supervise the UDF modeling." Requiring ground-truth bounding boxes and a curated mesh database severely restricts the scalability of the method.
3. Moreover, the ground-truth bounding boxes are usually annotated for "known objects", which almost can be seen as solved problem. The most important thing in lidar world model is to modeling the "general objects", which provides motion information out of the known list. By using the ground-truth bounding boxes, the whole system design would not have the ability to model the unknown objects.
4.

---

> ### Author Rebuttal · Authors · 2026-03-31
>
> W1: Our motivation to decompose the 4D space into four 3D subspaces aims to find the optimal balance between unambiguous geometric modeling and computational efficiency, taking a $N^4$ spatial-temporal feature grid as example:
>
> A. The Ambiguity of 2D Planes: Projecting 4D dynamics directly onto 2D planes (with a complexity of $6N^2$, like K-Planes) crosses too many dimensions, causing severe feature entanglement and structural overlap for inherently sparse LiDAR data. While sufficient for volume rendering in NeRFs, these highly entangled 2D projections fail to provide the unambiguous geometric priors strictly required by generative diffusion models.
>
> B. The Intractability of 4D Voxels: Conversely, a direct 4D voxel grid ($\mathcal{O}(N^4)$) is computationally prohibitive. Deploying heavy generative networks (e.g., 3D/4D DiT) on dense 4D tensors for large-scale autonomous driving scenes leads to instant memory overflow and intractable training times.
>
> C. The Superior Efficiency of 3D Subspaces + NCR: Our design reduces the initial complexity from $\mathcal{O}(N^4)$ to $4N^3$, explicitly avoiding the catastrophic projection loss of 2D planes. More importantly, our Neural Contourlet Representation (NCR) explicitly extracts directional information to maximize redundancy removal with minimal information loss. Under a 3-level NCR compression, our generative latent complexity drastically shrinks from $4N^3$ to $4(N/2^3)^3 = \frac{1}{128}N^3$.
>
> Mathematically, $\frac{1}{128}N^3 < 6N^2$ holds strictly true for any resolution $N < 768$. Given that typical latent resolutions $N$ for diffusion models are 128 or 256, our compressed 3D representation is factually more computationally efficient than K-Planes' 2D approach. This allows us to encode significantly higher spatio-temporal resolutions under an identical parameter budget.
>
> W2: We would like to clarify that annotations and proxy meshes are not required to define the core scene representation. The framework models the scene from raw LiDAR sequences, while annotations/mesh priors are introduced only to compensate for severe occlusion in dynamic regions and to improve supervision quality for novel-view synthesis. In other words, they are used as an auxiliary mechanism to recover missing geometry, rather than as a prerequisite for the reconstruction pipeline itself. We agree that this should be stated more clearly, and we will revise the paper to distinguish the core modeling framework from these optional supervision aids. Importantly, these priors can in principle be replaced by generic point-cloud completion methods, making the framework compatible with a fully annotation-free setting, see more in Re.cEzD-W1&Q1.
>
> W3&Q1: We clarify that our modeling and generation processes are inherently class-agnostic. Insight for GT bounding boxes is to serve merely as an optional data augmentation step for known classes and can be entirely skipped (refer more in W2 and Re.cEzD-W1&Q1). Our framework strictly decouples the scene into static and dynamic components without utilizing any fine-grained semantic labels during scaffold encoding or generative diffusion. The dynamic branch uniformly reconstructs and generates all moving elements in the scene based solely on continuous spatial-temporal UDF variations. Consequently, our Neural Field (NF) naturally captures 'unknown objects' in the exact same manner as known components. Just as the static scaffold uniformly encodes diverse, unlabeled background elements, dynamic unknown objects can be similarly encoded into our dynamic representation. This unified design ensures that our framework intrinsically supports generalized object modeling.

---

> > ### Author Rebuttal · Reviewer_8t8k · 2026-04-01
> >
> > The authors resolved my concern about the computation cost. However, even though the authors say their framework works without GT bounding box, I still cannot get how they can build the static scene PC without GT object removing, and how to handle dynamic unknown objects which has no velocity information.
> > Now I tend to keep my score on the negative side.

---

> > > ### Author Response · Authors · 2026-04-02
> > >
> > > Thank you for your continued engagement, for acknowledging our resolution regarding computational costs, and for highlighting the importance of scalable deployment. We appreciate the opportunity to clarify our GT-free capabilities and our mechanism for handling unknown dynamic objects. We have provided a framework illustration figure at https://anonymous.4open.science/r/Hiera4D.
> > >
> > > P1. First, we would like to clarify that our core pipeline (Fig. Hiera4D(a), red box) is entirely class-agnostic. It strictly processes pure 4D spatial-temporal geometry. Similar to K-Planes (CVPR23)[1], our Neural Scaffolds naturally learn to decouple static and dynamic components across their respective branches without requiring semantic labels.
> > > We originally utilized Ground Truth (GT) as an auxiliary tool for two reasons: A. Compactness Control: Explicit separation allows independent compression rate tuning for static and dynamic fields. In a unified field, finding the optimal balance between compactness and quality is more difficult (see Re. cEzD-Q2). B. Geometry Enhancement: GT aids in segmenting and completing occluded geometry, acting as data augmentation to improve novel-view rendering quality.
> > >
> > > P2. As shown in Fig. Hiera4D(b), we provide a framework figure to deploy without GT while maintaining explicit separation, we substitute GT dependencies with mature point cloud preprocessing techniques:
> > >
> > > A. Static/Dynamic Separation: We utilize ERASOR [2], which isolates dynamic points based on the physical characteristic that urban moving objects inevitably contact the ground. It uses a "pseudo occupancy" Scan Ratio Test and Region-wise Ground Plane Fitting (R-GPF) to identify regions with significant temporal changes, cleanly separating dynamic foreground from static background.
> > >
> > > B. Velocity-Free Clustering: The isolated dynamic points are clustered frame-by-frame using DBSCAN [4]. Because urban dynamic objects (cars, pedestrians) naturally maintain safe physical and social distances (visualized in Fig. Hiera4D(d)), spatial clustering is highly effective. Crucially, because this is done within single time frames, no velocity information is required.
> > >
> > > C. Handling Unknown Objects: We assign labels to these clusters using bounding-box size heuristics. For hard samples that do not fit standard rules, we apply finer segmentation (CVC-IROS19 [3]) to re-cluster and re-evaluate. If a cluster still does not match known classes, it is labeled as an "unknown object." Unknown objects bypass point cloud completion and are kept as raw, partial point clouds. They are then combined with the completed labeled objects to undergo self-supervised dynamic UDF learning.
> > >
> > > Because each frame is processed independently to reconstruct pure geometry, explicit velocity information is not needed. The Scaffold branches ($D^1_{xyz}$ and $D^{2-4}_{t..}$) simply self-learn the spatial-temporal representations.
> > >
> > > P3. We also visualzie a GT-free architecture (Fig. Hiera4D(c)) to demonstrate that our framework is also fully compatible with a purely separation-free approach. By bypassing the object retrieval step and removing the $D^5_{xyz}$ branch (the yellow region in Fig. Hiera4D(a)), we model the entire scene as a continuous 4D spatio-temporal Neural Field, similar to K-planes.
> > >
> > > Thanks for your kind suggestions indeed, your insights regarding GT-free deployment are incredibly valuable and chart a clear path for scaling our framework to in-the-wild datasets. We will thoroughly document these pre-processing scripts in our code release. We welcome any further questions and invite you to also refer to our corresponding response to Reviewer cEzD.
> > >
> > > [1] K-planes: Explicit radiance fields in space, time, and appearance, CVPR2023
> > >
> > > [2] ERASOR: Egocentric ratio of pseudo occupancy-based dynamic object removal for static 3D point cloud map building, RAL2021
> > >
> > > [3] Curved-Voxel Clustering for Accurate Segmentation of 3D LiDAR Point Clouds with Real-Time Performance, IROS2019
> > >
> > > [4] dbscan: A density-based algorithm for discovering clusters in large spatial databases with noise, KDD1996

---

### Official Review · Reviewer_cEzD · 2026-03-10

**Soundness:** 2
**Presentation:** 1
**Significance:** 2
**Originality:** 3
**Overall Recommendation:** 3
**Confidence:** 3

**Summary:**

This paper proposes introduces a hiearchical generative framework for scalable 4D LiDAR scene generation. The method consists of three main components: 1) a dynamic unsigned-distance-based scaffold that jointly models static background and dynamic foreground geometry. 2) a Neural Contourlet Rrepresentation (NCR) that compresses the scaffold into a compact and direction-aware latent representation via laplacian pyramid decomposition. 3) a Hierarchical Recoupling Generation (HRG) that progressively synthesizes large-scale 3D scenes through coarse-to-fine generation.  Experiments on KITTI-360 and Waymo demonstrate improvements over existing baselins in both reconstruction and generation quality.

**Compliance With Llm Reviewing Policy:**

Affirmed.

**Final Justification:**

The rebuttal and additional responses clarify and strengthen the paper significantly. However, I believe the paper would require a major revision cycle for these changes to be properly assessed. Therefore, I maintain my original score.

**Key Questions For Authors:**

1. Can the proposed method work without access to ground-truth annotations and the pre-defined mesh database?
2. Can the authors provide an ablation comparing a unified scaffold (no static-dynamic separation) against the proposed decomposed scaffold?

**Limitations:**

In conclusion, this paper has three main limitations. First, the proposed method relies on ground-truth annotations and an external mesh database to construct the scaffold representation, and the sensitivity of the overall pipeline to annotation and database quality remains unclear. Second, despite reporting comparable inference times to some existing methods, the computational cost of the multi-stage framework appears substantial, and the absence of a comparison with an efficiency-focused baseline such as R2Flow makes it difficult to fairly evaluate the practical efficiency of the approach. Third, the writing requires improvement. The authors spend considerable space describing the pipeline of converting point cloud into scaffold representation, yet fail to sufficiently analyze why this design choice enables effective and explicit modeling of temporal cues, which is the core problem motivating the work. The method description should be restructured to maintain a clear and consistent focus on addressing the major problem formulated in the paper.

**Strengths And Weaknesses:**

Strengths:
- The paper addresses an important challenge of scalable 4D LiDAR generation with spatio-temporal coherence in autonomous driving domain.
- Combining UDF-based scaffolds with learning compact LiDAR representations for LiDAR generation is a novel and well-motivated design.
- The empirical results are strong and consistent.
- The block-chunk generation  strategy is a sensible solution for large-scale unbounded scene generation

Weaknesses:
- The method rely on ground-truth annotations for background-foreground separation and a pre-defined database to supervise UDF modeling. This is a pratical limitation that is insufficiently discussed in the paper. The paper should clearly specify what annotations are required during trianing time and how sensitive the method is to annotation quality.

- The comparison with LiDARCrafter[1] in Table 4 is not fair. That experiment was conducted on nuScenes dataset, which is different from what the paper uses (KITTI-360 and Waymo). Since different datasets have fundamentally different scene layouts, sensor configurations, and dynamic content distributions, cross-dataset comparisons are not appropriate for assessing LiDAR generation quality. The authors should make the point clear in the paper. Additionally, the representation type of LiDARCrafter is BEV, not Range.

- Table 4 omits the inference time of R2Flow[2], a method specifically designed for efficient LiDAR generation that is already included in Table 2. Since the paper highlights efficiency as a key advantage, this omission is notable. Even if HieraScaffold does not match R2Flow's speed, the authors should report this comparison and honestly discuss the computational overhead of the multi-stage pipeline.

-  Trianing cost is not reported in the paper.

- The writing quality of the paper needs to be improved. The sentences are sometimes too difficult to parse. I'm not saying it is bad, but a consistent and well-structured explanation will improve the technical comprehension significantly, especially when the method  is already complex. Additionally, there are a few typos and grammatical errors throughout the paper:
	- "agiants" -> "against" in Figure 3 caption
	- "feautre" -> "feature" at line 156
	- "witu" ->  "with" at line 237
	- "learble" -> "learnable" at line 207
	- "for this time" at line 142

- (Minor) The qualitative comparisons in Figure 3 are difficult to interpret, as the visual differences between methods are subtle at the displayed scale. Providing zoomed-in views of representative regions would make the comparison more informative.


References:

[1]: Liang, A., Liu, Y., Yang, Y., Lu, D., Li, L., Kong, L.,
Zhao, H., and Ooi, W. T
 “LiDARCrafter: Dynamic 4D World Modeling from LiDAR Sequences.”
Proceedings of the AAAI Conference on Artificial Intelligence (AAAI), 2026

[2]: Nakashima, K., Liu, X., Miyawaki, T., Iwashita, Y., and Kurazume,
R. Fast lidar data generation with rectified flows.
In 2025 IEEE International Conference on Robotics and
Automation (ICRA), pp. 10057–10063. IEEE, 2025.

---

> ### Author Rebuttal · Authors · 2026-03-31
>
> W1&Q1: We thank the reviewer for highlighting this practical aspect. Our method functions without ground-truth (GT) annotations or predefined mesh databases. Currently, training uses object annotations (bounding box sizes and semantic classes) solely to initialize and accelerate occluded object completion via proxy meshes. This step is fully replaceable by annotation-free schemes (e.g., ICP algorithms or learning-based point cloud completion) without altering our core framework.
>
> A. Annotation Role: Bounding box annotations and object classes strictly initialize a multi-scaled trimmed ICP algorithm. This accelerates convergence and avoids local minima during mesh alignment to sparse LiDAR sequences. Pre-defined meshes provide complete geometry for heavily occluded regions, enabling comprehensive novel view sampling beyond the ego-vehicle's trajectory.
>
> B. Insensitivity to Annotation Quality: A new ablation study varying Gaussian noise on initialization bounding boxes demonstrates our framework's robustness:
> | Noise level | 0%    | 10%      | 20%     | 30%    |50%     |
> | -------- | -------- | -------- |-------- |--------|--------|
> | CD       | **0.0946**     | 0.0954     | 0.0951  | 0.0958 | 0.1023   |
> | F-Score  | **0.9214**     | 0.9208     | 0.9209  | 0.9203 | 0.9155   |
>
> Annotations merely crop coarse sub-point clouds. Our robust matching process ensures even 50% noise perturbation causes only minor scale shifts, yielding a marginal CD drop (~0.0077) from isolated failures rather than systemic breakdown. This proves high-precision annotations are unnecessary.
>
> C. GT/Mesh-Free Alternative: Our framework isn't bound to GT annotations or databases. We implemented an automated pipeline: 1. ERASOR (RAL '21) separates dynamic foregrounds without GT. 2. Point-Tr (ICCV '21) completes partial point clouds (replacing mesh queries), allowing direct UDF reconstruction via our static pipeline. Despite higher overhead and slightly reduced UDF quality in severe occlusions, it yields comparable metrics (CD/EMD: 0.0949/0.9212), as dataset evaluations prioritize ego-vehicle views where mesh-based occlusion-filling has minimal impact. We will detail this, include the ablation, and release the GT-free code.
>
> W2: The Controllable 4D Generation experiment was exclusively conducted on nuScenes to ensure a strict intra-dataset comparison with LiDARCrafter, rather than a cross-dataset comparison. We will clarify this setup and correct LiDARCrafter’s representation to "BEV" in Table 4.
>
> W3: We will add R2Flow to Table 4 (Representation: Range, JSD: 6.14, MMD: 0.67, Time: 0.2s). While R2Flow's Rectified Flow ensures faster overall inference (0.2s) via fewer steps, our single-step speeds are comparable. Crucially, our memory efficiency (25.1 vs. 77.8 GFlops) uniquely enables large-scale 4D generation on consumer GPUs.
>
> W4: Training takes ~3.5 days on 8x RTX 4090 (24GB). The coarse diffusion model trains for 1M steps (batch 128); the fine detail predictor for 0.6M steps (batch 32). All models use 1024 denoising steps for training, 256 for inference.
>
> W5&6: We will thoroughly proofread the manuscript to improve readability and update Figure 3 with zoomed-in views to better highlight qualitative differences.
>
> Q2: To demonstrate the necessity of static-dynamic separation, a new ablation study compares our decomposed scaffold against a joint "Unified Scaffold" baseline. Assuming a latent feature grid size of $256 \times 256 \times 32 \times 32$, we evaluate both the decomposed representation (denoted as $x/y$-NCR, where $x$ and $y$ indicate static and dynamic compression levels) and the unified representation (Uni/$x$-NCR). The parameters, reconstruction quality, and inference times are reported below:
> | Version | Params | CD | F-Score| inference time (s)|
> | -------- | -------- | -------- | -------- | -------- |
> | K-Planes    | 99.3k     | 0.1244     | 0.9040 | 3.87 |
> | 1/1-NCR    | 589.8k     | 0.0931     | 0.9236 | OOM |
> | 2/2-NCR    | 73.7k     | 0.0934     | 0.9229 | 3.43|
> | 2/3-NCR    | 37.8k     | 0.0946     | 0.9214 | 2.3 |
> | Uni/1-NCR    | 327.7k     | 0.0945     | 0.9210 | OOM |
> | Uni/2-NCR    | 40.9k     | 0.0987     | 0.9184 | 2.5 |
> | Uni/3-NCR    | 5.1k     | 0.1649     | 0.8982 | 0.5 |
>
> This demonstrates why static-dynamic separation is critical:
>
> A. Distinct Redundancies: Static backgrounds and dynamic objects possess inherently different spatiotemporal redundancies.
>
> B. Unified Scaffold Flaws: Uniform compression heavily biases the network toward the vast static background. Aggressive unified compression (e.g., Uni/3-NCR) indiscriminately destroys dynamic details, spiking CD from 0.0987 to 0.1649.
>
> C. Separation Advantages: Decomposition enables independent redundancy control. Customized compression (e.g., our optimal 2/3-NCR) aggressively prunes the static background while preserving dynamic details, optimally balancing efficiency (37.8k params), quality (0.0946 CD), and speed (2.3s) without OOM issues.

---

> > ### Author Rebuttal · Reviewer_cEzD · 2026-04-01
> >
> > Thanks to the authors for their detailed rebuttal. I acknowledge that several concerns have been adequately addressed. However, my primary remaining concern is the GT-reliance of the method. As far as I understand, the authors propose using ERASOR for background-foreground separation and Point-Tr for completing partial point clouds as a replacement for mesh queries. However, it is unclear how individual foreground instances are isolated and categorized without bounding box annotations or semantic labels.

---

> > > ### Author Response · Authors · 2026-04-02
> > >
> > > Thank you for your continued engagement and the opportunity to further clarify our GT-free framework. We have provided a framework illustration at https://anonymous.4open.science/r/Hiera4D.
> > >
> > > P1. Label-Free Core Pipeline.
> > > Our framework encodes and generates all foreground instances uniformly within the continuous spatial-temporal scaffold decomposition. As shown in the top red box of Fig. Hiera4D(a), this process is entirely class-agnostic and requires no semantic labels. During inference, our model predicts pure geometry, not semantic classes. Similar to 2D K-Planes (CVPR23), it naturally learns the static-dynamic separation process without additional annotations. The yellow region represents an optional annotation-guided enhancement step to complete occluded dynamic geometry.
> > >
> > > P2. GT-Free Framework for Scalable Deployment.
> > > Figure Hiera4D(b) details our pipeline for GT-free data. Our key insight is that off-the-shelf segmentation and point cloud preprocessing methods can achieve the necessary geometric enhancement without GT:
> > >
> > > A. We first apply the ERASOR[1] algorithm to separate the dynamic foreground from the static background.
> > >
> > > B. The static background is processed by our neural field as usual. For the unlabeled dynamic points, we adopt a cluster-then-complete strategy.
> > >
> > > C. As shown in the red box, we use DBSCAN to isolate most dynamic instances, exploiting the natural safe driving distances present in outdoor scenes. We then assign categories using heuristic bounding box size checks (which easily distinguish between cars, pedestrians, and cyclists). For hard samples, such as closely adjacent pedestrians (visualized in Fig. Hiera4D(d)), we apply a finer segmentation model (CVC[2]) and repeat the size check. Only successfully categorized instances are sent to Point-Tr for completion; unrecognized clusters bypass this step.
> > >
> > > D. Finally, all dynamic instances (both enhanced and bypassed) are combined for self-supervised dynamic UDF learning. As emphasized in point 1, 4D scaffold encoding requires only spatial-temporal geometric UDFs, completely bypassing the need for semantic labels.
> > >
> > > We greatly appreciate your insightful point regarding the necessity of a clear GT-free pipeline, which is indeed crucial for real-world deployment and wider adoption. While instance isolation serves primarily as a modular preprocessing step in our framework, allowing our core designs (the 4D Scaffold, NCR, and HRG generation in the red box of Fig. Hiera4D(a)) to remain purely geometry-driven, we fully agree that making this entire pipeline completely GT-free is practically significant. To ensure our 4D framework is fully scalable to unlabeled data, we will thoroughly document and release all corresponding preprocessing scripts and clustering details alongside our core codebase.
> > >
> > > [1] ERASOR: Egocentric ratio of pseudo occupancy-based dynamic object removal for static 3D point cloud map building, RAL2021
> > >
> > > [2] Curved-Voxel Clustering for Accurate Segmentation of 3D LiDAR Point Clouds with Real-Time Performance, IROS2019

---

### Official Review · Reviewer_8qQh · 2026-03-13

**Soundness:** 2
**Presentation:** 3
**Significance:** 3
**Originality:** 2
**Overall Recommendation:** 4
**Confidence:** 3

**Summary:**

This paper proposes HieraScaffold, a hierarchical representation and generation framework for 4D LiDAR scene generation. The method introduces a dynamic scaffold representation that models spatio-temporal geometry using unsigned distance fields, and a contourlet-based decoupling mechanism to obtain compact latent features. A hierarchical recoupling process then progressively reconstructs static scenes and dynamic objects to generate coherent 4D LiDAR sequences. Experiments on KITTI-360 and Waymo show improvements in reconstruction quality and temporal consistency over prior LiDAR generation approaches.

**Compliance With Llm Reviewing Policy:**

Affirmed.

**Final Justification:**

Thanks for the authors' rebuttal. I keep the original score unchanged.

**Key Questions For Authors:**

How sensitive is the model to the design choices of the scaffold hierarchy and contourlet decomposition?
Can the framework generalize to other LiDAR datasets or different sensor configurations?

**Limitations:**

The approach is evaluated mainly on two autonomous driving datasets and focuses on LiDAR scene generation. It remains unclear how well the method would scale to more diverse environments or other types of 3D dynamic data.

**Strengths And Weaknesses:**

++
The hierarchical scaffold representation is an interesting way to model dynamic LiDAR scenes.
The separation of static and dynamic components helps improve temporal consistency.
Experiments on large-scale datasets show promising results.

--
The method introduces several components (scaffold, contourlet decomposition, recoupling), making the framework somewhat complex.
The novelty relative to existing hierarchical or coarse-to-fine LiDAR generation methods could be clarified more clearly.
Some experimental comparisons and ablations could be expanded.

---

> ### Author Rebuttal · Authors · 2026-03-31
>
> W1-Clarification on Novelty: Current hierarchical and coarse-to-fine LiDAR generation methods (e.g., LiDM, UltraLiDAR, LaLaLiDAR) build explicit mappings on sparse point clouds, largely ignoring continuous neural representations. Conversely, our tightly coupled modules mutually reinforce each other to resolve the fundamental trade-off between structural fidelity and computational feasibility in 4D generation:
>
> A. Direct 4D Spatiotemporal Scaffold: Unlike methods that project data into 2D representations (e.g., K-Planes or Range images) and suffer information loss, our framework learns directly from raw LiDAR sequences to construct a continuous 4D scaffold, explicitly preserving rich 3D geometry and temporal dynamics.
>
> B. Neural Contourlet Representation (NCR) for Extreme Compactness: Existing hierarchical LiDAR generation methods (e.g., LiDM, UltraLiDAR) focus primarily on coarse2fine spatial mapping, ignoring inherent structural and frequency redundancy. To make generative modeling of this 4D scaffold tractable, NCR learns an exceptionally compact latent representation by aggressively pruning spatiotemporal redundancy in the frequency domain, enabling highly efficient diffusion training.
>
> C. Hierarchical Recoupling for Full World Generation: Existing hierarchical methods (e.g., LaLaLiDAR-AAAI26, LiDARCrafter-AAAI26) typically optimize structural cues to generate sparse point observations. In contrast, our framework generates the continuous 4D dynamic world itself. We introduce a hierarchical recoupling mechanism to progressively and efficiently recover high-fidelity, complete 4D scene details from the compacted latent space.
> These components create a mathematically rigorous pipeline enabling unbounded 4D scene generation on consumer-grade hardware (8x RTX 4090 24GB GPUs)—a feat dense baselines cannot achieve without enterprise GPUs (e.g., 8x A100 80GB).
>
> W1-More experimental comparisons and ablations: We added results including: Scaffold hierarchy level experiments (See Q1); Scaffold decoupling and NCR efficiency and performance reports (Re.cEzD Q2); annotation perturbation quality assessments(Re.cEzD W1Q1); and complete recoupling results(Re.523G W3).
>
> Q1-Sensitivity to Design Choices: Our framework's performance sensitivity to these design choices highlights their necessity.
>
> A. Scaffold Hierarchy: Implemented via $L=3$ levels of learnable multi-resolution feature grids at each node. As Table 5 demonstrates, removing these node features degrades Chamfer Distance by 11.6% (0.3214 to 0.3587), proving this structure is crucial for capturing fine-grained geometric fidelity. Additional results varying hierarchy levels while keeping the number of features constant are presented below:
> | Hierarchy Level | 1 | 2 | 3|4|
> | -------- | -------- | -------- |-------- |-------- |
> | CD     | 0.3587  | 0.3342 | **0.3214**    |0.3367|
> |F-Score |  0.8401 |  0.8507   | **0.8521**| 0.8501|
>
> Results indicate that increasing hierarchy levels generally improves geometric fidelity via multi-scale encoding (CD improves from 0.3587 at $L=1$ to 0.3214 at $L=3$). However, $L=4$ causes a performance drop. Because total parameters remain constant, excessive levels severely restrict feature allocation per level, hindering representational capacity. Nonetheless, the sub-optimal $L=4$ setting (CD: 0.3367) remains competitive, significantly outperforming the K-Planes baseline (CD: 0.4107, Table 1). Thus, $L=3$ is our optimal configuration. Furthermore, an added ablation (Unified vs. Decomposed Scaffold, Re.cEzD Q2) shows removing static-dynamic hierarchical separation causes catastrophic information loss for moving objects (CD worsens from 0.0946 to 0.1649).
>
> B. Contourlet Decomposition: As Table 6 demonstrates, decomposition filter choice significantly impacts performance. Replacing our learnable Neural filter with fixed filters (e.g., Haar or Biorthogonal) degrades FRD by up to 58.9%. Furthermore, removing the Directional Filter Bank (DB) noticeably reduces geometric fidelity, proving our specifically designed NCR is highly optimal for LiDAR's structural characteristics.
>
> Q2-Generalization to Other Datasets & Sensor Configurations: Our setup is unaffected by varying datasets and sensor configurations because we model directly in continuous 3D/4D space. Unlike range-view methods compressing data into 2D grids tied to specific LiDARs, our scaffold learns a continuous Unsigned Distance Field (UDF). Consequently, the underlying generated world is inherently sensor-agnostic. We simulate different sensor configurations by adjusting raycasting parameters (elevation angles, azimuth spacing, field of view) during final sampling. This allows tailored point cloud rendering for various real-world sensors using the same continuous scene. Alongside KITTI-360 and Waymo experiments, successful nuScenes evaluations further validate robust generalizability (see Table 4 for quality and inference time).

---

### Decision · Program_Chairs · 2026-04-30

**Decision:**

Accept (regular)

**Comment:**

Four reviewers gave overall positive scores: three ‘Weak accept’, and one ‘Weak reject’. During the rebuttal phase, the authors successfully addressed almost all the concerns raised by the reviewers. Based on all of these, the decision is to recommend the paper for acceptance. However, it is recommended that the authors revise the paper according to the reviewers’ comments when the paper is finally accepted.